



# Wake Characteristics of a Balloon Wind Turbine and Aerodynamic Analysis of its Balloon Using the LES-AD Model

Aref Ehteshami and Mostafa Varmazyar

Department of Mechanical Engineering, Shahid Rajaee Teacher Training University, Tehran, 1678815811, Iran

*Correspondence to*: Mostafa Varmazyar (varmazyar.mostafa@sru.ac.ir)

**Abstract:** Wake characteristics and aerodynamics of a balloon wind turbine were numerically investigated for different wind scenarios. Large eddy simulation, along with the actuator disk model, was employed to predict the wake behavior of the turbine. To improve the accuracy of the simulation results, a structured grid was generated and refined by using an algorithm to resolve about 80% of the local turbulent kinetic energy in the wake. Results

contributed to designing an optimized layout of wind farms and stability analysis of such systems. The capabilities of the hybrid ADM-LES model when using the mesh generation algorithm were evaluated against the experimental data on a smaller wind turbine. The assessment revealed a good agreement between numerical and experimental results. While a weakened rotor wake was observed at the distance of 22.5 diameters downstream of the balloon turbine, the balloon wake disappeared at about 0.6 of that distance in all the wind scenarios. Vortices generated by the rotor and

balloon started to merge at the tilt angle of $10°$, which intensified the turbulence intensity at 10 diameters downstream of the turbine for the wind speeds of 7 m s$^{-1}$ and 10 m s$^{-1}$. By increasing the tilt angle, the lift force on the wings experienced a sharper increase with respect to that of the whole balloon, which signified a controlling system requirement for balancing such an extra lift force.

## 1 Introduction

The global development of wind farms has decreased the area of potential land for building new wind power plants. Meanwhile, wind streams near the ground are not permanent or stable due to surface roughness and obstacles. To overcome these limitations, airborne wind energy (AWE) technology has been proposed as a way for harvesting wind power at higher altitudes (Paulig et al., 2013; Terink et al., 2011; Williams et al., 2007; Ruiterkamp and Sieberling, 2013; Günes et al., 2019; Cherubini et al., 2015). Compared to the conventional wind turbines at the 80-m altitude,

estimations have revealed that wind power density will be 2-9 times higher at the altitude of 400-1000 m, respectively (Ahrens et al., 2013). Taking this into account, it is possible to benefit from relatively stable and continuous wind. Moreover, there is no need to occupy vast land areas, and install towers for mounting wind turbines (Archer and Caldeira, 2009).

According to (Marvel et al., 2013), the maximum kinetic power that could be harvested from wind streams near the

Earth's surface (extracted from conventional wind turbines) was estimated at 400 TW, while it is 1800 TW at 200-1000 m altitudes in the entire Earth. Many mechanical systems have so far been introduced in line with the development of airborne technology (Paulig et al., 2013; Terink et al., 2011; Williams et al., 2007; Ruiterkamp and





Sieberling, 2013; Günes et al., 2019; Cherubini et al., 2015). For example, high-altitude kites (Paulig et al., 2013; Terink et al., 2011; Williams et al., 2007), gliders (Ruiterkamp and Sieberling, 2013; Sieberling, 2013), and Magnus effect turbines (Günes et al., 2019) are various types of airborne wind energy systems (AWESs). Among these, the balloon wind turbine, also known as the buoyant airborne turbine (Altaeros, 2022), has relatively simpler take-off and landing maneuvers due to the buoyancy effect. In a balloon wind turbine, the mechanical power of the rotor at the height of the balloon position, 400-1000 m, is converted into electrical energy by rotating the generator inside the balloon. This electrical energy is transferred to the ground by tethers, which are also responsible for controlling the balloon at the desired height.

The balloon of these turbines acts like a restricting duct that can raise wind speed, thereby increasing the rate of energy extraction from the wind (Bontempo and Manna, 2016; Van Bussel, 2007; Hansen et al., 2000; Lewis et al., 1977). Accordingly, Inglis (1979) corrected the power limitation of wind turbines obtained by the Betz theory (Betz, 1926) by considering the wake rotation and the mixture of the wake with the free-stream. Rauh and Seelert (1984) claimed that the output power obtained in the Betz theory did not lead to the accurate results. Accordingly, when calculating this power by the Betz theory, the pressure at the inlet and outlet of the control volume was assumed to be the same, while such an assumption was inconsistent with the pressure drop at the turbine location. Considering the pressure variations in this control volume, a modified equation for calculating the power was introduced.

Several studies have been conducted on ducted wind turbines (DWT) (Ahmadi Asl et al., 2017; Bontempo and Manna, 2016; Dighe et al., 2020; Lin and Porté-Agel, 2019; Tang et al., 2018; Lewis et al., 1977; Hansen et al., 2000; Van Bussel, 2007). Among them, experimental and theoretical studies by Lewis et al. (1977) exhibited an increase in DWT's power by a coefficient compared to bare turbines. They proved this coefficient equaled the total thrust of the duct and turbine to the turbine thrust alone. Van Bussel (2007) presented the improved momentum theory to investigate the effects of the duct on rotor performance. The mass flow rate of the air passing the duct was found to be directly related to the ratio of the duct's outlet area to its throat area.

Saeed and Kim (2016) were the first to investigate the unsteady behavior of a helium-filled buoyant balloon without a rotor by employing computational fluid dynamics (CFD) and fluid-structure interaction (FSI) simulations. In another study (Saeed and Kim, 2017), they added a scaled NREL Phase IV rotor to the buoyant balloon and predicted the aerodynamic performance of the rotor at different wind speeds and yaw angles by using the K-omega SST model. Their results showed that the presence of the balloon can increase the output torque by 17% compared to the bare turbine for the wind speed of 7 m s$^{-1}$. Recently, Ali and Kim (2020) attempted to consider the effects of wind shear, tilt, and yaw angle on the aerodynamic performance of a stand-alone rotor under wind conditions at 400-m altitudes and proved that both unsteady blade element momentum (UBEM) and CFD methods were reliable for calculating aerodynamic loads on rotor blades in airborne configuration, while the maximum amount of discrepancy between the methods did not exceed 4%.

In all previous numerical studies on balloon wind turbines (Saeed and Kim, 2017, 2016; Ali and Kim, 2020), the K-omega SST turbulence model, a time-averaged model which assumes steady and homogeneous turbulence, has been adopted. In contrast, the high-fidelity large eddy simulation (LES) model has been widely used in conventional wind turbine wake studies to resolve the transient behavior of wake flows more accurately (Lin and Porté-Agel, 2019; Porté-





Agel et al., 2011; Mo et al., 2013b, a). LES can resolve the large-scale turbulent motions that transport kinetic energy and momentum in wind turbine wakes. These large-scale motions are difficult to capture with RANS simulations, which are better suited for smaller-scale turbulence. This is particularly important for wind turbine wake simulations, where the large-scale structures dominate the flow behavior (Churchfield et al., 2012; Yang and Sotiropoulos, 2013). To resolve a wide range of turbulent motions using LES, it is necessary to use a fine mesh in regions of high turbulence,

such as the boundary layers and wake regions, (Pope, 2001). Hence, providing the required mesh for the rotor geometry and its boundary layer significantly increases total cell numbers and raises computational costs. On the other hand, using the actuator disk model (ADM) enables representing the turbine forces on airflow, needless to include the rotor geometry. Combining LES and ADM (LES-ADM) has emerged as a promising approach for predicting wind turbine wakes (Porté-agel, 2011; Yang and Sotiropoulos, 2013; Stevens et al., 2018) which can capture unsteady flow

features, such as vortex shedding and turbulent eddies, that are difficult to simulate using URANS-ADM. (Purohit et al., 2021) compared the LES-ADM and URANS-ADM for prediction of offshore wake losses against experimental results, they revealed that using LES-ADM in the wake simulation account for the more accurate forecast of turbulence intensity levels and velocity deficit in the wake region.

The LES-ADM is especially effective in predicting the flow variables in far-wakes, where the flow is less turbulent

and more uniform. Since, the near-wake areas are characterized by large eddies, which can considerably raise the turbulence intensity of the flow, they are not considered suitable for placing subsequent arrays of wind turbines. Therefore, accurate prediction of far-wake behavior is critical for optimizing wind farms. In this regard, Lin and Porté-Agel (2019) studied the turbine wake characteristics using a hybrid ADM and LES model for an incoming wind with a yaw angle. Their result revealed an acceptable agreement with both wind-tunnel measurements and analytical wake

models regarding wake deflections and spanwise profiles of the mean velocity deficit and turbulence intensity. According to previous studies, in this paper, the hybrid ADM-LES model based on the blade element momentum (BEM) theory is employed to predict the wake behavior of a balloon wind turbine and the aerodynamic forces acting on its balloon. Investigating the wake length behind these kinds of turbines is valuable as functioning in upstream turbines' wake flow means lower incoming wind speed, which leads to power losses and raises the oscillating loads

on downstream rotor blades, shortening their lifetime (Porté-agel, 2011).

Moreover, balloon wind turbines are considered non-crosswind systems among AWESs (Cherubini et al., 2015), and should be suspended at the specific altitude with preferred minimum displacement to perform in design conditions. The tethers attached to the ballon are responsible for its suspension and should balance the aerodynamic loads on the ballon. Consequently, studying the magnitude and behavior of the aerodynamic forces applied to the balloon in various

wind conditions is essential to dynamics analysis and control issues. Thus, the results of this investigation can be applied to promote the efficiency of balloon wind turbine farms in an optimized layout and design a controlling system for them. Additionally, a criterion for adjusting the grid size in the wake region and around the balloon was used in this research to resolve 80% of turbulent kinetic energy (TKE) in the wake directly in LES. Such a criterion has not been utilized to evaluate the grid size in the numerical study of the turbine wake so far (Porté-agel, 2011; Porté-Agel

et al., 2011; Lin and Porté-Agel, 2019; Mo et al., 2013b, a).





## 2 Computational model

Turbulent flows are characterized by unsteady behaviors and movement of eddies in a wide range of time and length scales. Theoretically, a turbulence model called direct numerical simulation (DNS) can directly resolve the entire spectrum of these eddies. However, this model demands high computational resources, especially for large domains and complex flows. Hence, different turbulence models have been developed by spatial averaging (Germano, 2000) and time averaging (Alfonsi, 2009) of Navier-Stokes equations to make an optimal trade-off between computational cost and accuracy. In LES, large eddies are resolved directly, and small eddies are filtered and modeled using secondary equations (Germano, 2000). Consequently, with respect to DNS, LES makes it possible to use a coarser mesh with a larger time step than DNS. In LES, the spatially-filtered continuum and Navier–Stokes equations are as follows:

$$\frac{\partial \rho}{\partial t} + \frac{\partial}{\partial x}(\rho \bar{u}_i) = 0 \tag{1}$$

$$\frac{\partial}{\partial t}(\rho \bar{u}_i) + \frac{\partial}{\partial x_j}(\rho \bar{u}_i \bar{u}_j) = \frac{\partial}{\partial x_j}(\sigma_{ij}) - \frac{\partial \bar{p}}{\partial x_i} - \frac{\partial \tau_{ij}}{\partial x_j} \tag{2}$$

where $\bar{u}_i$ is the resolved velocity in the i-direction (i = 1, 2, and 3 correspond to the x, y, and z directions), $\bar{p}$ is the filtered static pressure, $\sigma_{ij}$ is the stress tensor due to molecular viscosity, and $\tau_{ij}$ is the subgrid-scale stress, defined by the following equations:

$$\sigma_{ij} = \left[\mu\left(\frac{\partial \bar{u}_i}{\partial x_j} + \frac{\partial \bar{u}_j}{\partial x_i}\right)\right] - \frac{2}{3}\mu \frac{\partial \bar{u}_i}{\partial x_i}\delta_{ij} \tag{3}$$

$$\tau_{ij} = \rho \overline{u_i u_j} - \rho \bar{u}_i \bar{u}_j \tag{4}$$

The subgrid-scale stresses calculated from the filtering operation are unknown and require modeling. The subgrid-scale turbulence models use the Boussinesq hypothesis (Manwell et al., 2010) and the subgrid-scale turbulent stresses can be obtained from the following expression:

$$\tau_{ij} - \frac{1}{3}\tau_{kk}\delta_{ij} = -2\mu_t \bar{S}_{ij}, \tag{5}$$

where $\mu_t$ is the subgrid-scale turbulent viscosity. The isotropic part of the subgrid-scale stresses $\tau_{kk}$ is not modeled, but added to the filtered static pressure term. The rate of strain tensor for the resolved scale is indicated by $\bar{S}_{ij}$ and can be calculated from the following:

$$\bar{S}_{ij} = \frac{1}{2}\left(\frac{\partial \bar{u}_i}{\partial x_j} + \frac{\partial \bar{u}_j}{\partial x_i}\right) \tag{6}$$

Germano et al. (1991) and, subsequently, Lilly (1992) have conceived a procedure, in which the Smagorinsky model constant, $C_S$ , is dynamically computed based on the information provided by the resolved scales of motion. The governing equations and assumptions used in this model can be obtained from (Kim, 2004). Since dynamic Smagorinsky-Lilly model exhibited accurate results in the wake study of conventional wind turbines (Lin and Porté-





Agel, 2019; Mo et al., 2013b; Porté-Agel et al., 2011), it was employed to predict the wake characteristics of balloon
        wind turbines in the current study.

## 3 Wind turbine modeling

The actuator disk model was used to model the rotor in this study. In this approach, the rotor is considered a rotating
disk in the flow field and the reaction forces of the rotor on air are obtained by employing the BEM. The rotating or
actuator disk (AD) is the area swept by the turbine blades. The BEM calculates the mutual forces between the blade
and the air by combining the momentum and blade element theory. In the following sections, these two methods are
briefly explained and, in the last section, the turbine's and the balloon's geometry are described.

### 3.1 Momentum theory

The schematic views of the control volume enclosing the AD and an annular element at the rotor plane are illustrated
in Figure 1. In this figure, r, U, and $a$ represent element radius, air velocity, and the axial induction factor, respectively.
The axial induction factor is the fractional difference in wind velocity between the free-stream and the rotor plane.

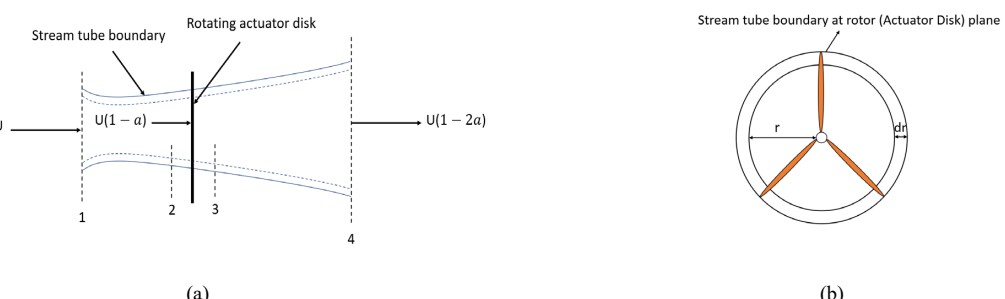

(a)                                                                 (b)

**Figure 1. (a) Control volume, (b) Annular element**

Applying the conservation of linear and angular momentum to the control volume and the annular element, thrust, $dT$
can be computed by Eq. (7), and torque, $dQ$ exerted on this element can be calculated from Eq. (8).

$$p_2 - p_3 = \rho(\Omega + \tfrac{1}{2}\omega)\omega r^2, \tag{7}$$

$$dQ = d\dot{m}(\omega r)r = (\rho U_2 2\pi r dr)(\omega r)r \tag{8}$$

where $\rho$, $P_2$, and $P_3$ represent air density and the static pressure just before and after the rotor, respectively. Moreover,
$U_2$, $\Omega$, $\omega$ and $\dot{m}$ indicate the air velocity just before the rotor, rotor angular velocity, the angular velocity imparted to
the flow stream, and the mass flow rate passing through the annular element, respectively. Using the definition of
induction factor in Eqs. (7) and (8), $dT$ and $dQ$ can be obtained from Eqs. (9) and (10), respectively. In Eq. (10), $\acute{a}$ is
the angular induction factor ($\acute{a} = \omega/2\Omega$ ).

$$dT = \tfrac{4a(1-a)}{2}\rho U^2 2\pi r dr \tag{9}$$





$$dQ = 4\acute{a}(1-a)\rho U \pi r^3 \Omega dr \tag{10}$$

An assumption used in the momentum theory is the infinite number of blades. To consider the finite number of blades in calculating aerodynamic forces on the blade, Prandtl's tip loss factor, F, can be used as defined in Eqs. (11) and

(12) (Hansen, 2015). Additionally, the simple momentum theory breaks down when the axial induction factor is greater than 0.4. In this regard, the Glauert correction, which is an empirical relation between the thrust coefficient, $C_T$, and $a$, can be used as defined in Eq. (13) to comply with measurements (Hansen, 2015). In the following equations, R is the rotor radius and $a_c = 0.2$.

$$F = \frac{2}{\pi} \cos^{-1}(e^{-f}), \tag{11}$$

$$f = \frac{B}{2} \frac{R-r}{r \sin \varphi}, \tag{12}$$

$$C_T = \begin{cases} 4a(1-a)F & a \leq a_c \\ 4(a_c^2 + (1-2a_c)a)F & a > a_c \end{cases} \tag{13}$$

### 3.2 Blade element theory

In this approach, each rotor's blade is divided into N elements, as depicted in Figure 2 (a). Each element can be generated from a separate airfoil. Aerodynamic forces on each element are calculated based on the lift coefficient, $C_l$,

drag coefficient, $C_d$, chord length, c, and angle of attack (AOA). Relative wind is the vector sum of wind velocity at the rotor, $U(1-a)$ and the wind velocity due to the rotation of the blade, $\Omega r(1+a')$. This rotational component is the vector sum of the blade section velocity, $\Omega r$, and the induced angular velocity at the blades, $\omega r/2$, from the conservation of angular momentum. The relation between various forces, angles, and velocities at the blade is shown in Figure 2 (b) looking down from the blade tip.

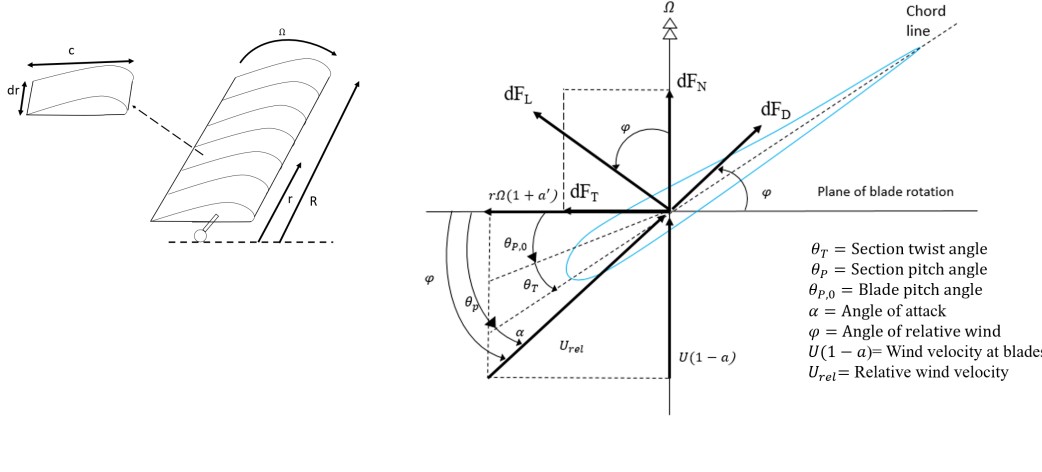

(a)                           (b)



**Figure 2. (a) Schematic of blade elements; dr, radial length of element; r, radius, (b) airfoil geometry for analysis of forces on a horizontal axis wind turbine.**

In Figure 2 (b), $dF_L$ is the lift force, $dF_D$ is the drag force, $dF_N$ is the force normal to the plane of rotation (which contributes to the thrust), and $dF_T$ is the force tangent to the circle swept by the rotor. Considering a rotor with B blades, the total normal force, $dF_N$, on a section with radius $r$ and thickness $dr$, can be computed from Eq. (14) using geometric relations between different forces in Figure 2. Moreover, the total torque, $dQ$, exerted on this section can be estimated by multiplying $dF_T$ by $r$ as Eq. (15) :

$$dF_N = B \frac{1}{2} \rho U_{rel}^2 (C_l \cos\varphi + C_d \sin\varphi)cdr \qquad (14)$$

$$dQ = B \frac{1}{2} \rho U_{rel}^2 (C_l \sin\varphi - C_d \cos\varphi)crdr \qquad (15)$$

According to Figure 2 (b), the relative wind velocity, $U_{rel}$ can be computed using Eq. (16). Therefore, Eqs. (14) and (15) can be written as Eqs. (17) and (18).

$$U_{rel} = \frac{U(1-a)}{\sin\varphi} \qquad (16)$$

$$dF_N = \acute{\sigma}\pi\rho \frac{U^2(1-a)^2}{\sin^2\varphi}(C_l \cos\varphi + C_d \sin\varphi)rdr, \qquad (17)$$

$$dQ = \acute{\sigma}\pi\rho \frac{U^2(1-a)^2}{\sin^2\varphi}(C_l \sin\varphi - C_d \cos\varphi)r^2 dr \qquad (18)$$

where $\lambda_r = \frac{r\Omega}{U}$ and $\acute{\sigma} = \frac{Bc}{2\pi r}$ are the local speed ratio and solidity, respectively.

When calculating induction factors in BEM, the accepted practice is to set $C_d$ to zero (Wilson and Lissaman, 1974). For airfoils with low drag coefficients, this simplification introduces negligible errors. By equating normal force (Eqs. 185   (9) and (17)) and torque (Eqs. (10) and (18)) equations from the momentum and blade element theory, and after some algebraic manipulation, Eqs. (19)-(23) are gained. Glauert correction and Prandtl tip loss factor are considered in defining these equations.

$$C_l = 4F \sin\varphi \frac{(\cos\varphi - \lambda_r \sin\varphi)}{\acute{\sigma}(\sin\varphi + \lambda_r \cos\varphi)} \qquad (19)$$

$$\acute{a} = \frac{1}{\frac{4F\cos\varphi}{\acute{\sigma}C_l} - 1}, \qquad (20)$$

$$a = \frac{1}{\frac{4F\sin^2\varphi}{\acute{\sigma}C_l\cos\varphi} + 1} \qquad a \leq a_c, \qquad (21)$$

$$a = \frac{1}{2}\left[2 + K(1 - 2a_c) - \sqrt{(K(1 - 2a_c) + 2)^2 + 4(Ka_c^2 - 1)}\right] \qquad a > a_c \quad , \qquad (22)$$

$$K = \frac{4F\sin^2\varphi}{\acute{\sigma}C_l} \qquad (23)$$

Since $\varphi = \theta_p + \alpha$, for a given blade geometry and operating conditions, there are two unknowns in Eq. (19), namely $C_l$ and $\alpha$ per section. To find these values, one can use the empirical $C_l$ versus $\alpha$ curves of the airfoil (Vries, 1979).

One then finds $C_l$ and $\alpha$ from the empirical data satisfying Eq (19). Once $C_l$ and $\alpha$ are known, $\acute{a}$ and $a$ can be



determined from Eqs. (20)-(23). After finding these values, the axial force and torque exerted on the blade section (equal to reacting forces on wind streams) can be obtained from Eqs. (17) and (18).

### 3.3 Turbine geometry

In this study, a turbine with 3-m diameter, d, and 0.75-m clearance from the inner surface of the balloon was used. The rotor comprises two blades and rotates around its horizontal axis with angular velocity $\Omega = 251$ rpm. The S809 airfoil, used in the NREL Phase VI rotor's blades, was selected for all blade sections. This rotor has shown good aerodynamic performance in wind turbine applications (Hand et al., 2001; Simms et al., 2001). Empirical curves of $C_l$ and $C_d$ versus $\alpha$ for this airfoil can be extracted from (Airfoil Tools, 2022). To calculate the optimized value of $\theta_P$ per section, the QBlade software was employed. This software detects the AOA, at which $C_l/C_d$ is optimal for a given airfoil and operating condition, and then uses this value to calculate the optimized $\theta_P$ per section. Radial variations of the chord length and pitch angle are presented in Table 1.

**Table 1**
**Radial variation of the pitch angle and chord length for the turbine**

| Radius(m) | 0.12 | 0.19 | 0.25 | 0.4 | 0.52 | 0.63 | 0.8 | 1.2 | 1.5 |
|---|---|---|---|---|---|---|---|---|---|
| Pitch(degree) | 11.24 | 6.67 | 3.83 | 1.9 | 1.5 | 1.2 | 1 | 0.8 | 0.5 |
| Chord (m) | 0.07 | 0.14 | 0.11 | 0.093 | 0.08 | 0.071 | 0.063 | 0.057 | 0.052 |

To model the effects of the hub on wind streams, a cylinder with the radius of 0.08 m and drag coefficient of $C_{d,hub} = 1.2$ was used according to (Churchfield et al., 2015). The axial aerodynamic force exerted from the hub can be calculated from Eq. (24). In this equation, $R_{hub}$ and $U_{hub}$ represent the hub radius and average wind speed at the hub height, respectively. After calculating all the aerodynamic forces by using AD and hub properties (Eqs. (17), (18), and (24)), they were employed as sources of momentum in the domain by developing user-defined function (UDF) codes in ANSYS-FLUENT.

$$F_{hub} = \frac{1}{2}\rho\pi R_{hub}^2 U_{hub}^2 C_{d,hub} \tag{24}$$

### 3.4 Balloon geometry

A schematic representation of the turbine's balloon is shown in Figure 3 (a). The balloon consists of a duct and four wings and is suspended by tethers. NACA 9414 airfoil was used in the duct cross-section to increase the wind speed passing the turbine (Saleem and Kim, 2018). To improve the stability of the balloon in strong winds, four wings with n0009sm-il airfoil cross-sections were attached to the duct (Saeed and Kim, 2016). The geometric dimensions of the balloon and the actuator disk (AD) position inside it are presented in Figure 3 (b). As noted in this figure, the AD was located in the duct's throat to experience higher wind speed.





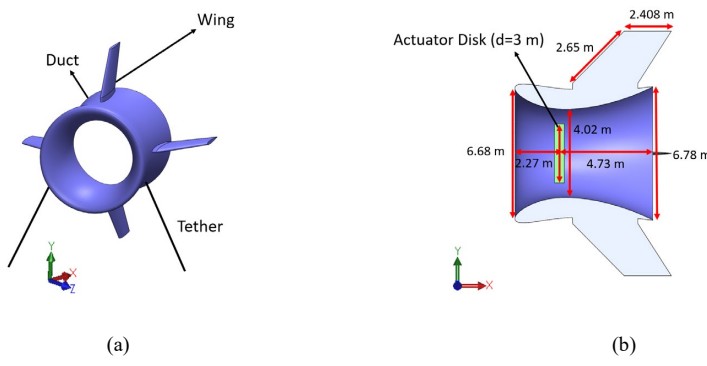

(a)
(b)

**Figure 3. (a) the 3D model of the balloon with a duct (NACA 9415 airfoil cross-section) and 4 wings (n0009sm-il airfoil cross-section), (b) Geomatic dimensions of the balloon and AD position inside it.**

## 4 Computational domain

Yaw ($\theta_{yaw}$) and tilt ($\theta_{tilt}$) angles represent the angle between the rotor axis and its horizontal and vertical axes, respectively (Figure 4). Since the balloon and rotor are symmetric with respect to x-y and x-z planes (Figure 3), yawed and tilted configurations are technically equivalent. Therefore, only the effects of tilt angle on the wake flow were considered in this numerical investigation.

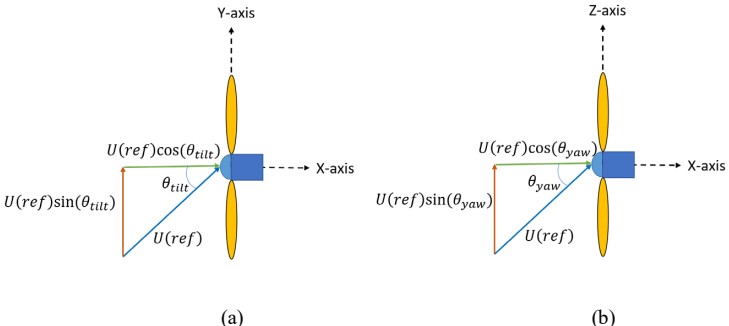

(a)
(b)

**Figure 4. (a) tilt configuration (b) yaw configuration**

A cuboid domain was created around the balloon in the current study. To ensure zero gradients of flow variables on the boundaries, the upstream length, downstream length, and width of the cuboid domain were determined as 5, 22.5, and 15 times of the turbines' diameter, respectively. These dimensions are illustrated in Figure 5. Unsteady simulations of the balloon wind turbine were performed with inlet reference velocity, $U_{ref} = 7$ m s$^{-1}$, 10 m s$^{-1}$, and $\theta_{tilt} = 0°$, 5° and 15° to compare the behavior of the wake in various wind flow scenarios. For the non-tiled cases, only one inlet along the x-axis was considered. However, in the tilted cases, a secondary flow was entered into the domain along the y-axis to ensure the full exposure of the balloon wind turbine to the tilted wind stream (Figure 5 (b)). The velocity boundary condition was employed at the inlet(s) of the domain, while the pressure boundary condition was specified



at the outlet(s). No-slip and stress-free walls were assigned to the balloon body and side walls of the domain,
respectively. For all the wind scenarios in this study, the altitude of 400-m was assumed. The corresponding air density
and viscosity at this altitude were set to $\rho = 1.179\ kg\ m^{-3}$ and $\mu = 1.8 \times 10^{-5} kg\ m^{-1}s^{-1}$, respectively.

The time step was set corresponding to the rotor rotation of $2.5°(\Delta t = 0.0016\ s)$. LES calculations were run sufficiently
to reach stable statistics of the flow. In other words, after passing this criterion, the time-averaged flow variables have
not experienced significant changes. On this basis, the simulation time was set at approximately 9.2 s, corresponding
to 40 rotations of the rotor. The 1333 sampling data during this time were taken to perform the time-averaging process
on the flow variables.

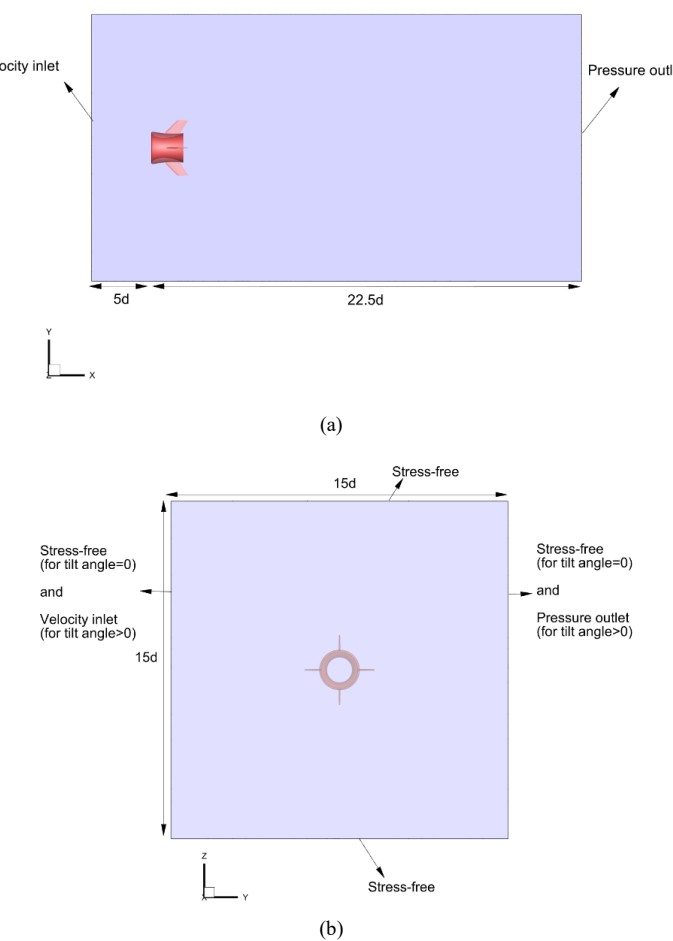

**Figure 5. Size of the computational domain and boundary conditions by looking at (a) x-y plane (b) z-y plane**



## 5 Grid generation

### 5.1 Mesh resolution in LES

LES requires mesh sizes sufficiently fine to resolve the energy-containing eddies. The mesh resolution determines the

fraction of TKE directly resolved. In this study, a method was adopted to generate a mesh that could directly resolve about 80% of the local TKE spectrum in the wake flow and the region enclosing the balloon. Moreover, the entire domain was discretized with hexahedral elements using ANSYS-ICEM. In the following, the concepts and equations of this approach are described.

According to Figure 6 (a), resolving an eddy with diameter $l$ requires 4 cells with the width of $l/2$ $(=\Delta_c)$. However, if

the width of two adjacent cells around the eddy is less than $l/2$, as shown in Figure 6 (b), this eddy cannot be resolved directly and the sub-grid scale models will be used to resolve it.

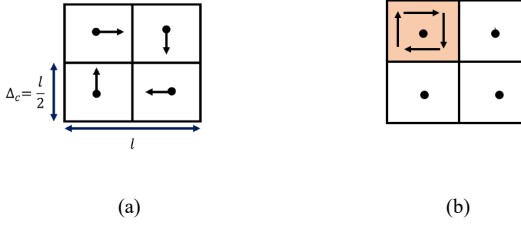

(a)                      (b)

**Figure 6. Cells size around a (a) resolved eddy, (b) modeled eddy**

A schematic view of the eddies downstream of the turbine in three zones (A, B, and C) and their corresponding turbulence energy cascade (TEC) diagrams are depicted in Figure 7. While zone A is located in the wake flow, zones C and B are far from the wake region. TEC diagrams in Figure 7 represent the relation between the wave number of

eddies, $w(=2\pi/l)$, and their kinetic energy density, $E$. According to these diagrams, as the length scale (diameter) of eddies decreases, their wave number $w$ increases while their kinetic energy density declines. Considering the location of zone A in the turbulent wake region, the length-scale of eddies and their kinetic energy density in this region are higher $(E_1>E_2)$ than those of zones B and C.

In Figure 7, more details of the TEC diagram in zone A are described. The area under the TEC curve indicates the

total TKE of all eddies in this zone. The smallest size of the resolved eddies is considered to be $l_1$. The area under the TKE curve for eddies smaller than $l_1$ (the dotted area) is modeled, while that of greater sizes (the crosshatched area) is directly resolved via LES.





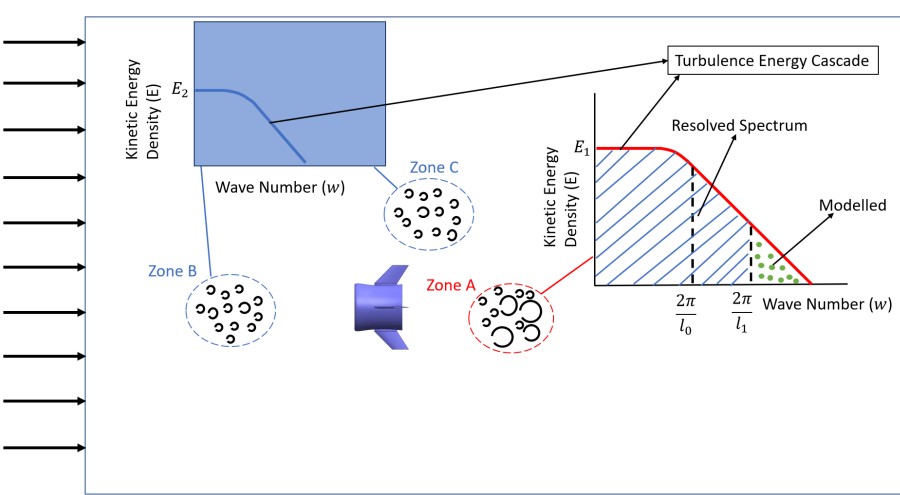

**Figure 7. Zone A, B, and C and the corresponding TEC diagrams**

All eddies in each arbitrary region can be equated with an eddy with $l_0$ length-scale called integral length-scale (Gerasimov, 2016). This eddy contains the averaged energy of all the eddies in the determined region, and its size can be computed from Eq. (25), in which $k$ represents the TKE of an eddy with wave number $w$.

$$\frac{l_o}{2\pi} = \frac{\int_0^\infty w^{-1} k(w) l(w)}{\int_0^\infty k(w) l(w)} \tag{25}$$

Table 2 presents the relation between the resolved fraction of total TKE (dotted area in Figure 7, $k(l_1)$), $l_1$, and cell size, $\Delta_c$, based on the Kolmogorov's energy spectrum (Gerasimov, 2016). Here, $l_0$ and $\Delta_c$ can be calculated from Eqs. (26) and (27). In these equations, $\varepsilon$ and V represent the dissipation rate of the TKE and cell volume, respectively. According to this table, to resolve about 80% of the total TKE in a region, there should be at least five cells along $l_0$. To resolve this amount of TKE in the wake flow and around the balloon, an algorithm was used to obtain the mesh size. Different steps of the algorithm are shown in Figure 8.

$$l_o = \frac{k^{\left(\frac{1}{3}\right)}}{\varepsilon} \tag{26}$$

$$\Delta_c = V^{\left(\frac{1}{3}\right)} \tag{27}$$

**Table 2**
**The relation between k ($l_1$), $l_1$ and $\Delta_c$ based on Kolmogorov's energy spectrum**

| Resolved fraction of total TKE | $\dfrac{l_1}{l_0}$ | $\dfrac{l_0}{\Delta_c}$ |
|---|---|---|
| $k(l_1) = 0.1\, k$ (10%) | 6.1 | 0.33 |
| $k(l_1) = 0.5\, k$ (50%) | 1.6 | 1.25 |
| $k(l_1) = 0.8\, k$ (80%) | 0.42 | 4.8 |
| $k(l_1) = 0.9\, k$ (90%) | 0.16 | 12.5 |




According to Figure 8, a coarse mesh was generated in ANSYS-ICEM at the beginning of each simulation. In the following step, a precursor RANS simulation was performed to estimate $l_0$ and $\Delta_c$ around the balloon and in the wake region from Eqs. (26) and (27). Next, the contours of $l_0/\Delta$ were plotted to check the mesh resolution in these regions. When $l_0/\Delta$ was smaller than 5 in specified regions, the first mesh was refined, and the process was continued from step 2; otherwise, the LES simulation was started using the first mesh.

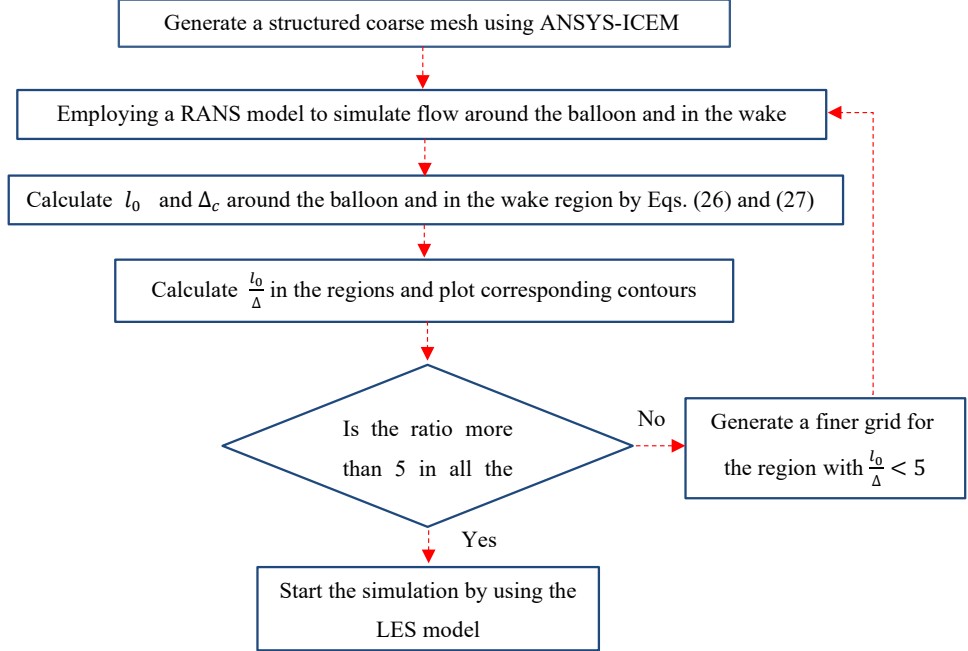

**Figure 8. The steps in generating mesh to resolve 80% of total TKE around the balloon and in the wake**

### 5.2 Blocking method and mesh independence study

To discretize the cuboid computational domain by hexahedral elements, it was divided into three regions along its axial axis (upstream, downstream, and chord length of the balloon), as shown in Figure 9. Next, the four O-grid topologies were used to split these blocks in the radial direction to control the mesh inside and in the width of the duct, wings, and the region around the balloon. To accurately capture the boundary layer effects, the value of y+ was set to approximately 1 and 20 inflation layers were assigned with the growth ratio of 1.2 in the direction normal to the walls. Mesh distribution in the domain using the criterion was named mesh G2 and its details are given in Table 3.

To further assess the criterion, its grid independence was investigated. Accordingly, two coarser (G1) and finer (G3) meshes, summarized in Table 3, were generated. Using these meshes in three simulations, the time-averaged pressure coefficients $C_P$ on a circle on the balloon (Figure 10 (a)) were compared. All of the simulations were performed for $U_{ref} = 10$ m s$^{-1}$ and $\theta_{tilt} = 0°$ and the results are illustrated in Figure 10 (b).





According to Figure 10 (b), the maximum value of the mean $C_P$ is related to the position of the wings due to their stagnation effect on the incoming flow. The difference between the peak $C_P$ in meshes G2 and G3 is about 2%, while this difference is around 7% for meshes G2 and G1. Since the difference between the results of meshes G2 and G1 is about one-third of the difference between those of meshes G2 and G3, using the mesh criterion in LES can satisfy the

295    mesh independence criterion. All the simulations were performed by high-performance computing (HPC) systems having GPUs to accelerate the calculation time.

**Table 3**
**Mesh distribution in the computational domain for evaluating mesh criterion in LES.**

| Grid number | G1 | G2 | G3 |
|---|---|---|---|
| Nodes on edge Nin | 23 | 43 | 50 |
| Nodes on edge Nwing | 25 | 35 | 40 |
| Nodes on edge Nout | 25 | 30 | 40 |
| Nodes on edge Nu | 20 | 30 | 40 |
| Nodes on edge Ns | 74 | 86 | 95 |
| Nodes on edge Nd | 250 | 285 | 300300 |
| Nodes on edge Np | 140 | 196 | 236 |
| Nodes within the boundary layers Nbl | 35 | 35 | 35 |
| Total Number of nodes Nt | 4,972,096 | 10,756,364 | 15,4657,804 |

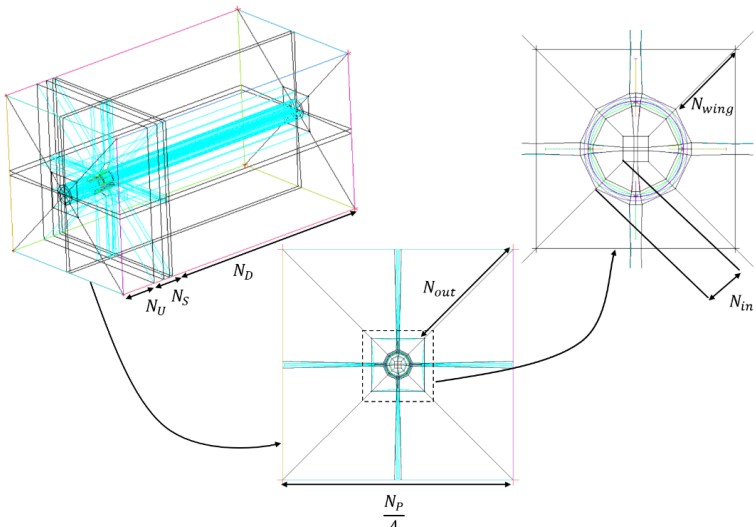

**Figure 9. Mesh topology around the balloon**

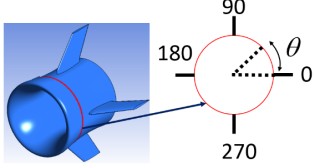

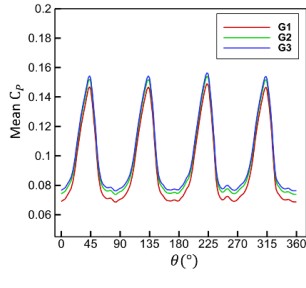

(a)                    (b)





**Figure 10. (a) the circle on the balloon, (b) the comparison of mean $C_P$ against $\theta$.**

Figure 11 depicts the mesh on the balloon; two perpendicular planes pass through it, with a magnified view of the boundary layer's mesh. Evidently, the boundary layers' mesh thoroughly covered the walls to improve the accuracy in calculating viscous forces.

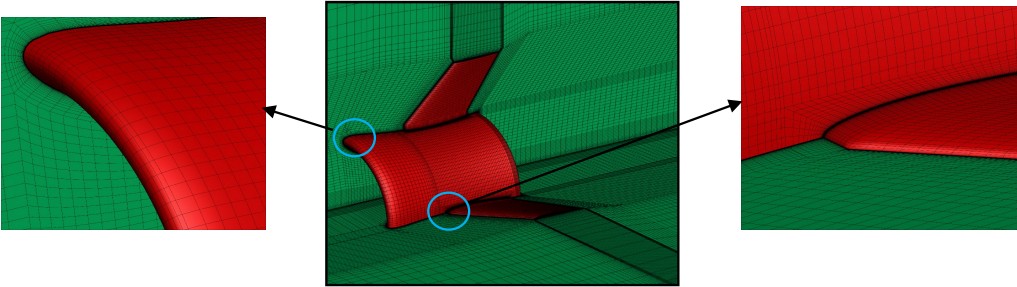

**Figure 11. Mesh distribution on the balloon body, two perpendicular planes passing through, and an exaggerated view of mesh in the boundary layers of one wing and the duct.**

## 6 Results and discussions

### 6.1 Validation of simulation

Since balloon wind turbine systems are novel research topics, experimental data regarding wake analysis were unavailable to validate the results of the current study. To assess the capabilities of the hybrid ADM-LES model and the mesh criterion in LES, the wake behavior of a smaller conventional turbine (functioning on the ground) was studied, which had been studied experimentally before (Chamorro and Porté-Agel, 2010). The corresponding turbine diameter and angular velocity were 1.5 cm and 1220 rpm, respectively. An unsteady simulation was run for 50 s flow-time to obtain stable time-averaged flow statistics. The position of AD and the location of one upwind line and seven downwind lines of the turbine (x/d = -1, 2, 3, 5, 7, 10, 14, 20) for 0 <y/d<2 at the symmetry plane of the channel (z=0) are illustrated in Figure 12. At the end of the simulation, the time-averaged streamwise velocity at these locations was calculated and compared with those from the experimental study in Figure 13.

In Figure 13, the boundary layer effects can be observed on the exponential upwind profile. Looking at downstream profiles in Figure 13, a good agreement between the experimental measurements and the ADM-LES method is observed for all the locations. The results showed that the numerical model can accurately predict the non-uniform velocity deficit in the near and far wake. Deviation of the numerical results from wind tunnel measurement at x/d=2 close to the ground can be justified since the tower effects were neglected in the numerical investigation. The maximum velocity deficit occurs at the hub height. Moreover, the wake effects can be seen until x/d=20, where the time-averaged streamwise velocity profile is at its primary shape due to momentum extraction from the free-stream.





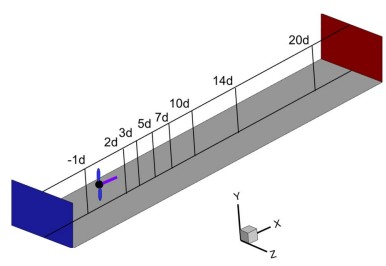

**Figure 12.** **The location of selected lines in upwind and downwind of the turbine.**

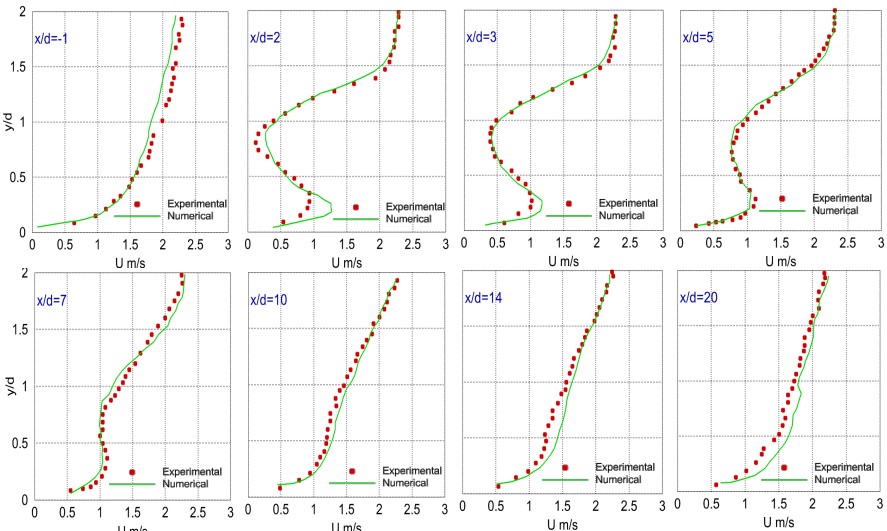

**Figure 13. Comparison of vertical profiles of the time-averaged streamwise velocity U (m s⁻¹): ADM-LES (solid line), wind-tunnel measurements (dotted line).**

### 6.2 Velocity profiles in the balloon wind turbine wake

In this section, the results of the wake analysis of the balloon wind turbine are reported and discussed. Figure 14 displays the vertical profiles of the time-averaged normalized x-velocity for one upstream and seven normalized downstream locations behind the turbine for $-4 < y/d < 4$ at z=0. For a better understanding of the wake behavior in different wind scenarios, various incoming velocities ($U_{ref} = 7$ m s⁻¹ and $10$ m s⁻¹) and different tilt angles ($\theta_{tilt} = 0°$, $5°$ and $10°$) were used. In Figure 14, the stagnation effects of the turbine on the incoming air are apparent on upwind

profiles in all the cases.

     These effects lead to the 1% velocity deficit for the regions $-2.1 < y/d < 2.1$, $-2.3 < y/d < 1.3$, and $-2.5 < y/d < 1.1$ at $\theta_{tilt} = 0°$, $5°$, and $10°$, respectively. On the other hand, there is clear evidence of velocity deficit in three regions just when



the airflow passes through the balloon. These regions contribute to rotor wake, wings wake, and separation effects in the trailing edge of the duct's airfoil. The wake structures in these three regions are investigated separately below.

According to Figure 14 (a), the normalized x-velocity profiles for $U_{ref} = 7$ m s$^{-1}$ are quite similar to $U_{ref} = 10$ m s$^{-1}$ everywhere in the wake. However, at $U_{ref} = 7$ m s$^{-1}$, relatively faster recovery can be observed in the rotor wake. It can be explained by the higher value of TSR for $U_{ref} = 7$ m s$^{-1}$ at the same rotational speed leading to increased shear turbulence in the wake, as observed in other studies (Krogstad and Adaramola, 2012; Martínez-Tossas et al., 2022). Higher shear turbulence in the wake can cause more lateral turbulent diffusion into the region and, consequently, faster

recovery (Krogstad and Adaramola, 2012; Martínez-Tossas et al., 2022). The difference in the recovery rate is noticeable up to x/d=10, where the difference in the averaged x-velocity in the wake width is maximized at 6% for the two wind speeds.

The effect of tilt angle on wake behavior is depicted in Figures 14 (b) and (c). Irrespective of the incoming wind speed, it can be observed that the wake is deflected toward the tilted direction as the tilt angle increases. Moreover, the wake

becomes asymmetric around the rotor axis compared with the non-tilted inflow, and the wake center (location of the minimum velocity) shifts upward in the lateral direction as the wake progresses further downstream. By increasing $\theta_{tilt}$, there is a reduction in the wake width of the rotor for locations far from x/d=7. This reduction can be attributed to two reasons. Firstly, by increasing the tilt angle, there is less interaction between the wind flow and the rotor because of a decrease in the rotor swept area, as seen from the incoming flow, which provides less momentum to be extracted

by the rotor. Secondly, as the tilt angle increases, there is an increment in the supply of momentum from the surrounding free-stream into the wake. As a result, the velocity deficit in the wake recovers faster. Hence, the time-averaged x-velocity in the wake width at x/d=22.5 reaches 0.93, 0.94, and 0.96 of the free-stream value for $\theta_{tilt} = 0°$, $5°$, and $10°$, respectively.

Comparing the results of tilted and non-tilted inflow in Figure 14, it can be interpreted that the effect of tilt angle on

momentum transformation into the wake is stronger than TSR as there is a slight difference in velocity profiles for different $U_{ref}$ in Figure 14 (b) and (c). The wake width reduction and deflection may be desirable in increasing the incoming wind speed of any downstream balloon wind turbine in wind farms. Therefore, tilt angle and TSR are essential factors in designing an optimized layout of wind farms for balloon wind turbines.

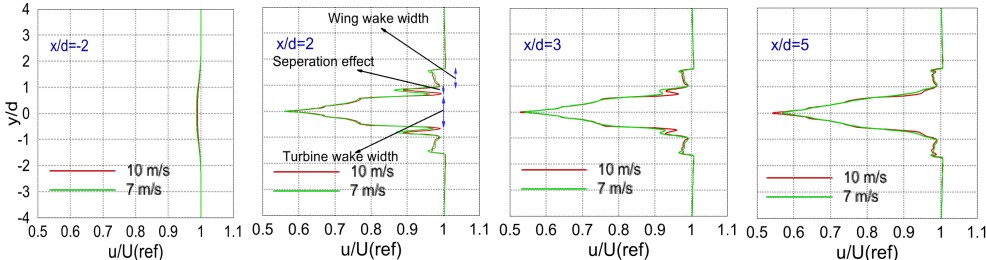





(a)

(b)

(c)



**Figure 14. Comparison of vertical profiles of the time-averaged normalized x-velocity for one upstream and seven downstream locations for -4<y/d<4 at z=0 in $\theta_{tilt}$= (a) 0$^\circ$ (b) 5$^\circ$ (c) 10$^\circ$**



Considering the velocity profiles just after the turbine (x/d=2) in Figure 14, there is evidence of velocity defect due to the presence of the wings. However, the wings' effect on the velocity field is not as significant as the effects of separation. It should be claimed that, in all wind scenarios investigated in this study, the wings did not experience the stall phenomenon and no separation was observed around them, which can justify their negligible effect on the velocity field. According to Figure 14, the wings' effect on flow disappears at distances as large as x/d=14 in all the cases. The third distinct region in the wake structure in Figure 14 is related to the separation effect in the trailing edge of the duct's airfoil. Figure 15 shows streamlines near the trailing edge of the duct's airfoil for U$_{ref}$= 7 m s$^{-1}$ and $\theta_{tilt}$ = 5$^\circ$ at the symmetry plane of the balloon (z=0). From this figure, the growing rate of the duct cross-section after the throat generates a positive pressure gradient in the streamwise direction, which leads to flow separation in the boundary layer of the duct's inner surface.

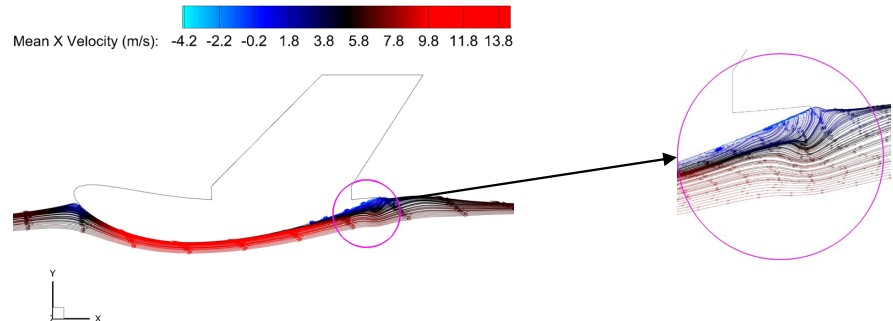

**Figure 15. Separated streamlines near the trailing edge of the duct's airfoil for U$_{ref}$ = 7 m s$^{-1}$ and $\theta_{tilt}$ = 5$^\circ$ at the symmetry plane of the balloon (z=0).**

### 6.3 Turbulence intensity profiles in the balloon wind turbine wake




Similar to conventional wind farms on the ground, turbulence intensity (TI= $u'/U$) is a key factor in studying wind farms of balloon wind turbines. From one viewpoint, high turbulence intensity in the airflow can amplify the frequency of aerodynamic loads on blades, which may adversely affect rotor dynamics and shorten the blade's lifetime (Ismaiel and Yoshida, 2018). On the other hand, when the frequency of the oscillations approaches the system's natural frequency, a resonance phenomenon may happen, which leads to high-amplitude oscillations and affects the balloon's stability. In this regard, vertical profiles of the time-averaged turbulence intensity for locations mentioned above in the previous wind scenarios are illustrated in Figure 16. As is evident from the figure, the presence of the balloon wind turbine does not increase TI in the incoming upstream flow. On the other hand, right after the flow passes through the balloon, it experiences a noticeable TI increment in two distinct regions. The first region's width is equal to the diameter of the rotor at -0.5<y/d<0.5. The TI growth in this region is related to large-scale vortices generated by the angular momentum transformation of the actuator disk, which moves downstream in the wake. The second region is





related to flow separation in the trailing edge of the duct's airfoil (Figure 15). Vortices in this region can promote TI along the duct's width (-0.8475 <y/d<-0.66) and (0.66<y/d<0.8475).

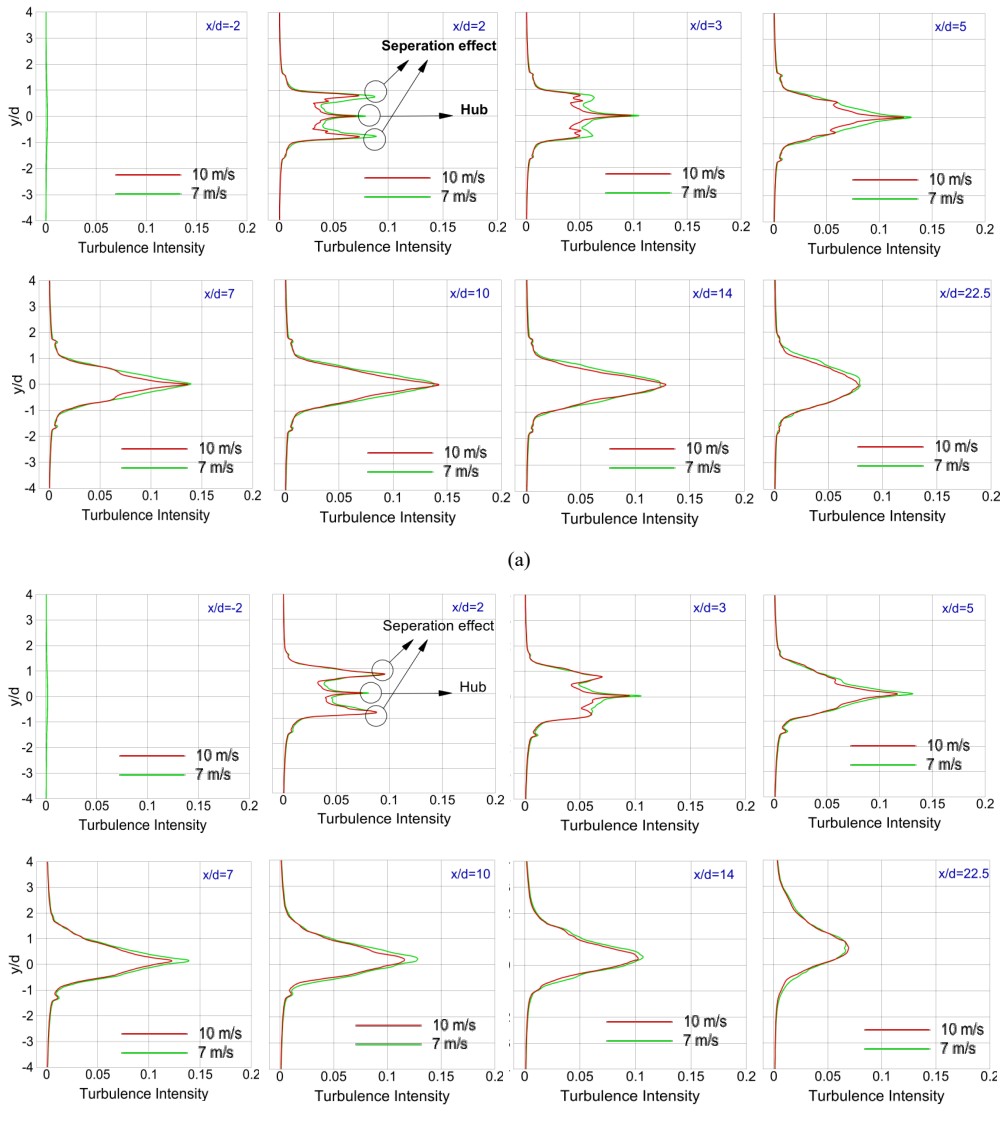





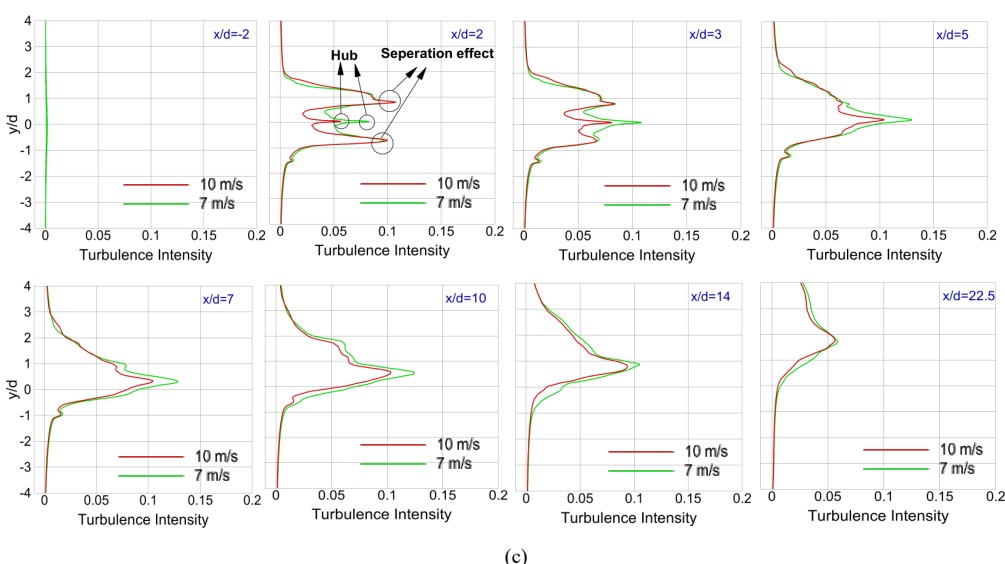

(c)

**Figure 16.** Comparison of vertical profiles of the time-averaged turbulence intensity for one upstream and seven downstream locations for -4<y/d<4 at z=0 in $\theta_{tilt}$= (a) 0° (b) 5° (c) 10°.

From Figure 16, higher TI can be observed at all the tilt angles at $U_{ref}$ = 7 m s$^{-1}$ and almost everywhere downstream of the turbine. This is due to higher shear turbulence in the wake of higher TSR cases, as described in the previous section. It is observed that in the case of $\theta_{tilt}=0°$ and $U_{ref}$ = 7 m s$^{-1}$, TI is higher than 10 m s$^{-1}$ until x/d=10, which can explain faster recovery for $U_{ref}$ = 7 m s$^{-1}$ up to x/d=10 as observed in Figure 14 (a). By increasing the tilt angle in Figure 16 (b) and (c), the TSR difference between two incoming wind speeds will grow with a coefficient of $(1/\cos{(\theta_{tilt})})$, which explains the greater TI difference than the non-tilted inflow.

From Figure 16, TI in the rotor wake (-0.5<y/d<0.5) for $\theta_{tilt}=0°, 5°$, and 10° gradually increases until x/d=10, 7, and 5, respectively, and then starts to fall. This reduction can be attributed to the instability of the wake flow and the breakdown of large-scale vortices (eddies) into smaller vortices in these locations. When large-scale eddies flow downstream in the wake, their energy will be transferred into smaller eddies; in the smallest scales, this energy is converted into the internal energy of the flow. The results show that the energy cascade or the transfer of energy from large-scale motions to small-scale ones in this region continues at distances as large as y/d=22.5. However, in the region attributed to separation effects, the energy cascade continues up to x/d=5 for $\theta_{tilt}=0°$ and 5° ; then, the maximum TI remains below 5% further downstream. Note that, at $\theta_{tilt}=10°$, vortices generated by the rotor and the separation are merged, and their effects are significant even at 10d downstream of the turbine.

### 6.4 Aerodynamic loads on the balloon

Figure 17 shows a schematic view of the resultant aerodynamic forces and the resultant torques exerted on the balloon. As shown in this figure, the system has 6 degrees of freedom, demanding complex control systems. Several tethers are connected from the base station to the balloon responsible for controlling the system. These tethers should be



connected at specific points and directions on the balloon to balance aerodynamic forces and prevent high increments of yaw and tilt angles. Studies on the control and dynamic model of the system are outside the scope of the current study and may be the subject of future research. In this section, aerodynamic forces acting on the balloon are numerically investigated by employing the LES-ADM method for the same wind scenarios used in previous sections. Total aerodynamic forces are divided into pressure forces and viscous forces. The magnitude of these forces in x (drag forces) and y directions (lift forces) are plotted in Figure 18. Considering no inflow along the z-axis from boundaries into the domain and the symmetrical geometry of the balloon about the x-y plane, the resultant force in this direction is nearly zero for all the cases ($R_Z = 0$).

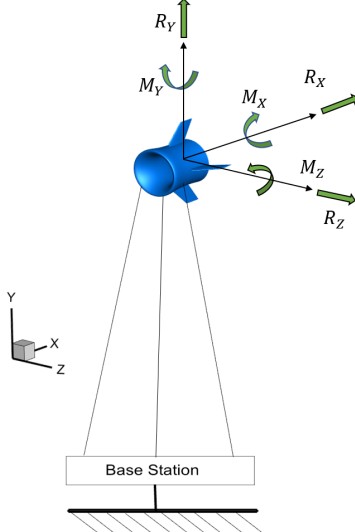

**Figure 17. A schematic of aerodynamic loads on the balloon**

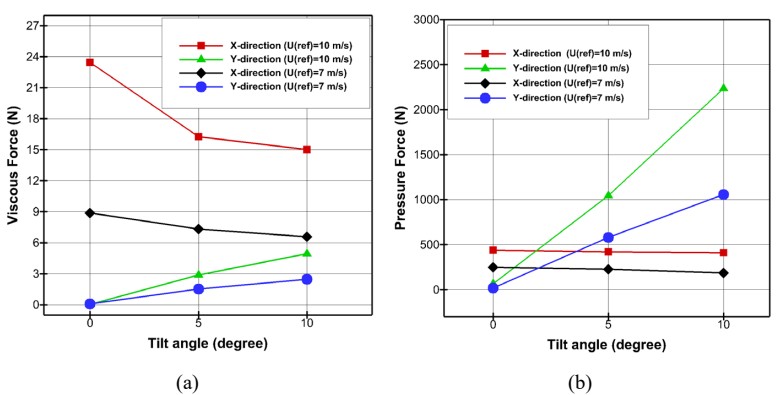

**Figure 18. (a) viscous forces and (b) pressure forces on balloon in the x and y directions variation with wind speeds and tilt angles**



From Figure 18 (a), it can be seen that with increasing $\theta_{tilt}$ to $10°$, viscous forces in the y direction increase marginally to 5 N for $U_{ref}$ = 10 m s$^{-1}$ and decrease to about 6 N for $U_{ref}$ = 7 m s$^{-1}$ in the x direction. This trend may be due to the increment of the y-component and reduction of the x-component of incoming flow when the tilt angle rises. As displayed in Figure 18 (b), pressure forces in the y direction have a direct relation with tilt angle and they reach 1055 N and 2235 N for $U_{ref}$ = 7 m s$^{-1}$ and 10 m s$^{-1}$, respectively, at $\theta_{tilt} = 10°$. On the other hand, the magnitude of the

pressure forces slightly decreases in the x direction when the tilt angle rises, but they always remain under 500 N in all wind scenarios. It should be pointed out that for tilted inflows, pressure forces are always higher in the y direction. The lower order of viscous forces relative to pressure forces was predictable and was approved by the results.

For a better understanding of the relation of these forces with tilt angle and wind speed, aerodynamic loads on the duct and wings should be analyzed separately. Figure 19 shows $C_l$ and $C_d$ versus the AOA (tilt angle in the current study)

for the wings' cross-section airfoil at $U_{ref}$ = 7 m s$^{-1}$. Since the airfoil has a symmetric profile, the corresponding lift coefficient is zero and the drag coefficient is very low (about 0.004) at $\theta_{tilt} = 0°$. According to the figure, a stall occurs when the AOA exceeds $13°$, which is above the maximum tilt angle in all the cases used in the current study.

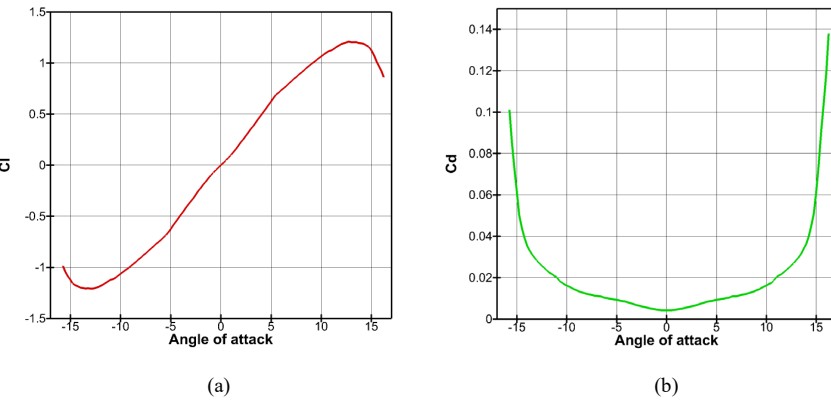

(a)             (b)

**Figure 19. (a) lift coefficient (b) drag coefficient versus angle of attack in 7 m s$^{-1}$ wind speed**

In Figure 19, all the wings are named and the magnitudes of aerodynamic forces acting on wing Wh1 are plotted in different directions. Regarding $\theta_{yaw} = 0°$, there is a symmetric velocity field around wings Wv1 and Wv2, which

generates very low magnitudes of lift and drag forces by considering Figure 19. Furthermore, due to the symmetric flow pattern about the x-y plane, vectors of resultant forces on wings Wh1 and Wh2 are equal. As wing Wh1 operates below the stall angle, the corresponding lift force experiences a rise as the tilt angle increases, which is evident from Figure 20 (b). In addition, the magnitude of the lift force at $\theta_{tilt} = 10°$ is 73 N for $U_{ref}$ = 7 m s$^{-1}$, and 146 N for 10 m s$^{-1}$. By comparing Figure 18 and Figure 20 (c), the portion of the lift forces exerted on the wings in the total lift force exerted on the balloon can be calculated. Hence, this value was measured to be around 12% and 14% for $U_{ref}$ = 7 m s$^{-1}$

$^{1}$ and 10 m s$^{-1}$, respectively, at $\theta_{tilt} = 5°$. At $\theta_{tilt} = 10°$, it experiences a rise being almost 28% for $U_{ref}$ = 7 m s$^{-1}$ and 29% for $U_{ref}$ = 10 m s$^{-1}$. It can be concluded that, by increasing the tilt angle, the ratio of the lift force on the wings to the total lift force on the balloon will increase; therefore, a controlling system is required for balancing such extra lift force.



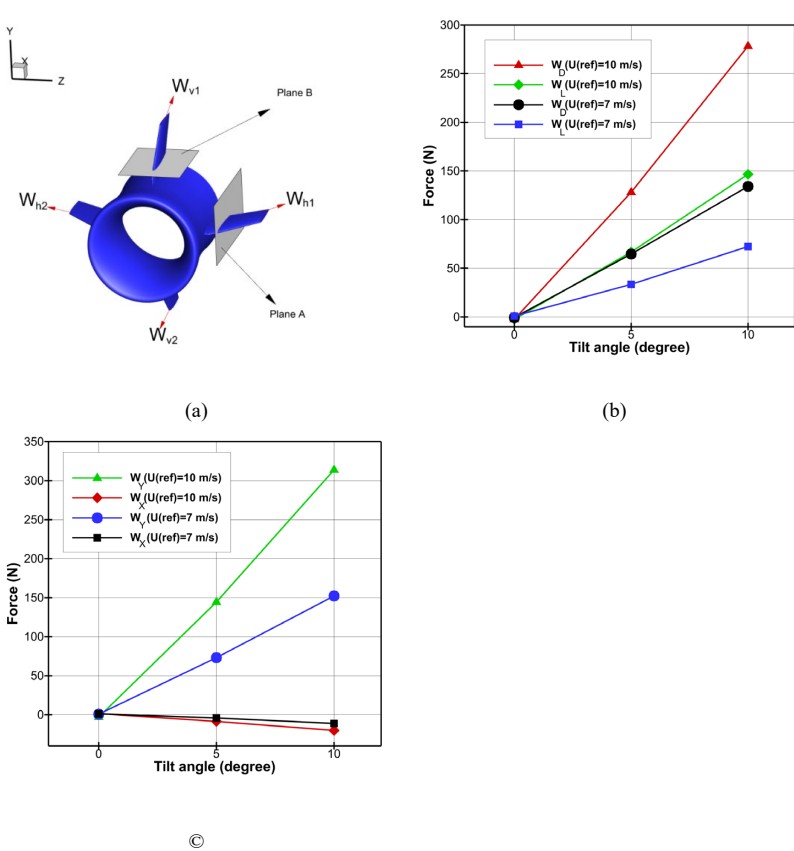

Figure 20. (a) wings' names (b) lift and drag forces on wing $W_{h1}$ (c) projection of the corresponding forces in the x and y directions in different wind scenarios

In Figure 21 (a), pressure contours on the inner and outer surfaces of the right surface of the duct are presented for $U_{ref} = 7$ m s$^{-1}$ and $\theta_{tilt} = 0°$. Taking into account the symmetric flow and geometry concerning the x-y plane, the resultant aerodynamic forces on both sides are equal in the y direction. As shown in the figure, the pressure on the outer side of the surface is higher than that of the inner side in different areas, except for the area near the leading edge. Therefore, the pressure difference generates a resultant force toward the center of the balloon. In Figure 21 (b), a schematic representation of these resultant forces is shown on two separate airfoils (A1 and A2) at different circumferential positions on the duct ($-\beta$ and $\beta$ around the x-axis). As the airfoils are symmetric about the x-z plane, aerodynamic forces exerted on them are equal in magnitude at $\theta_{tilt} = 0°$, $(F_{R2})_Y = (F_{R1})_Y$, but they act in opposite directions. Therefore, the total lift force on the right surface (and the lift surface) is zero in the non-tilted inflow. However, for the tilted inflows, the pressure distributions on the pressure side of these airfoils and on the suction side are not the same; consequently, $(F_{R2})_Y \neq (F_{R1})_Y$ which can generate a lift force on the duct.



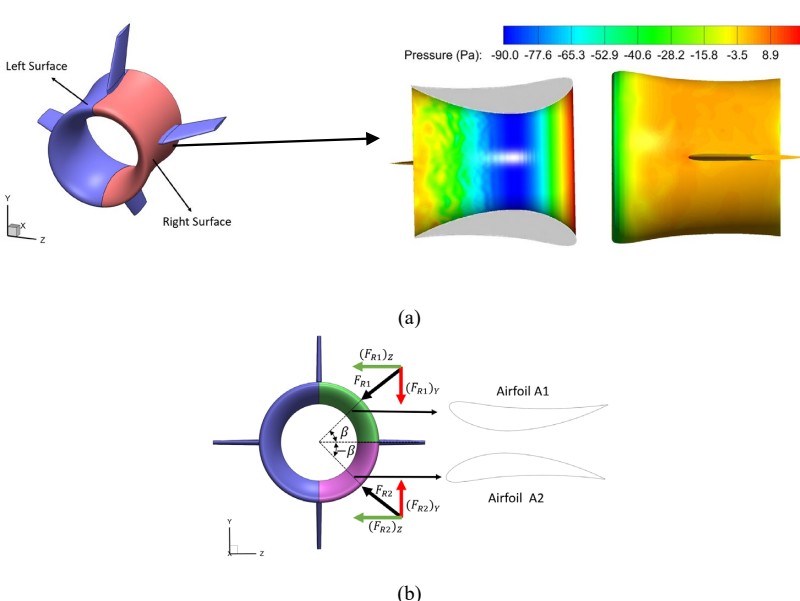

Figure 21. (a) pressure contour on the inner and outer surfaces of the right surface of the duct in $U_{ref}$ = 7 m s$^{-1}$ and $\theta_{tilt} = 0°$ (b) schematic of the resultant forces on two separate airfoils in different circumferential positions on the duct ($-\beta$ and $\beta$ around x-axis).

Figure 22 illustrates the time-averaged pressure coefficient for A1 and A2 airfoils in the streamwise direction at $\beta = 30°, 45°$, and $60°$. The mean $C_P$ in this figure was calculated at $U_{ref}$ = 7 m s$^{-1}$ and $\theta_{tilt} = 10°$. From the figure, there is an obvious difference in mean $C_P$ at each normalized distance for different values of $\beta$. This discrepancy is due to different pressure distributions on their pressure and suction sides for tilted inflows, as described before.

Consequently, for tilted inflows, $(F_{R2})_Y > (F_{R1})_Y$, and the balloon will experience an aerodynamic lift force in addition to the buoyant force, which can help the system float in the air.

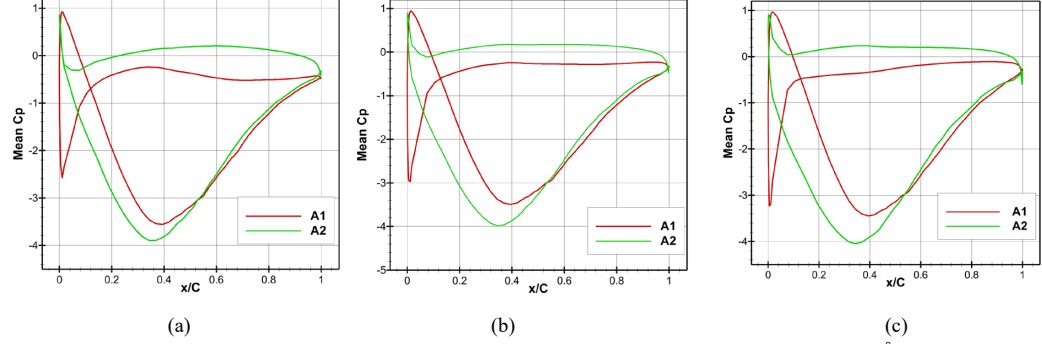

Figure 22. Mean $C_P$ for A1 and A2 airfoils in streamwise direction for $U_{ref}$ = 7 m s$^{-1}$ and $\theta_{tilt} = 10°$ in (a) $\beta = 30°$, (b) $\beta = 45°$ and (c) $\beta = 60°$.



## 7 Conclusion

The wake characteristics and aerodynamics of a balloon wind turbine were numerically investigated for different wind
scenarios by employing a hybrid LES-ADM model. A mesh criterion was used to resolve more than 80% of TKE in
the wake and around the balloon. The conclusions drawn from the results of this study are as follows:

1. The mesh generated by using the aforementioned criterion and the proposed algorithm satisfied mesh
   independence requirements.
2. The wake structure of the balloon wind turbine consisted of three distinct regions affected by the wings, rotor,
   and separation in the trailing edge of the duct's airfoil. For non-tilted inflows, higher TSR led to faster
   momentum recovery in the wake at distances as large as 10d from the turbine. However, by increasing the
   tilt angle, the effect of this parameter on the wake recovery was reduced and the wake shifted toward the
   tilted direction. It was concluded that both TSR and tilt angle are essential factors in designing an optimized
   layout of a balloon wind turbine farm.
3. When the airflow passed through the balloon, it experienced a rise in two distinct regions affected by the
   separation along the duct's width (-0.8475 <y/d<-0.66 and 0.66<y/d<0.8475) and the wake behind the rotor
   (-0.5<y/d<0.5). The TI in the rotor wake rose until 10d, 7d, and 5d downstream for $\theta_{tilt} = 0°, 5°, 10°$,
   respectively. In these locations, vortices generated by the rotor broke down into smaller eddies and,
   consequently, TI declined.
4. In the region corresponding to separation, energy transformation to smaller eddies continued up to 5d
   downstream for $\theta_{tilt} = 0°, 5°$, and the maximum TI in this region remained below 5% in further downstream
   locations. However, at $\theta_{tilt} = 10°$, eddies generated by the separation and rotor were merged and their effect
   remained significant at distances as large as 10d from the rotor.
5. An upward trend was observed for the lift force versus tilt angle such that, at $\theta_{tilt} = 10°$, it reached 1055 N
   and 2235 N for incoming wind speeds of 7 m s$^{-1}$ and 10 m s$^{-1}$, respectively. Furthermore, as the tilt angle
   increased, the portion of the wings' lift in the total balloon's lift reached 29% at $\theta_{tilt} = 10°$, which was
   recognized as an important factor in the controlling system design of the balloon.
6.

## CRediT authorship contribution statement

**Aref Ehteshami:** Methodology, Simulation & Validation, Formal analysis, Writing - Original Draft preparation.
**Mostafa Varmazyar:** Supervision, Writing - Review & Editing

## Declaration of competing interest

The authors declare that they have no conflict of interest.

## Acknowledgments



The authors acknowledge the computing resources provided by Shahid Rajaee Teacher Training University.

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
