# Peer review of "Wake Characteristics of a Balloon Wind Turbine and Aerodynamic Analysis of its Balloon Using the LES-AD Model"

_Wind Energy Science, 2023_

## Author Comment (AC1)

Response to Anonymous Referee #1

Authors' responses to reviewer comments appear in blue text. Line numbers referenced in the authors' responses refer to the revised document. Figures with Arabic numerals (e.g., Figure 5) correspond to the revised manuscript; figures with Roman numerals (e.g., Figure vi) only appear in response to the reviewer's comments.

We sincerely thank the reviewer for their valuable comments, which have greatly improved our manuscript.

1. Little bit of background is required in the abstract;

Thank you for your comment regarding the background information in the abstract. In the updated abstract, we have provided more details about airborne wind energy systems and particularly balloon wind turbines, as below:

In the realm of novel technologies for generating electricity from renewable resources, an emerging category of wind energy converters called Airborne Wind Energy Systems (AWESs) has gained prominence. These pioneering systems employ tethered wings or aircraft that operate at higher atmospheric layers, enabling them to harness wind speeds surpassing conventional wind turbines' capabilities. The balloon wind turbine is one type of AWESs that utilizes the buoyancy effect to elevate the turbine to altitudes typically ranging from 400 to 1000 meters. In this paper, the wake characteristics and aerodynamics of a balloon wind turbine were numerically investigated for different wind scenarios. Large eddy simulation, along with the actuator disk model, was employed to predict the wake behavior of the turbine. To improve the accuracy of the simulation results, a structured grid was generated and refined by using an algorithm to resolve about 80% of the local turbulent kinetic energy in the wake. Results contributed to designing an optimized layout of wind farms and stability analysis of such systems. The capabilities of the hybrid ADM-LES model when using the mesh generation algorithm were evaluated against the experimental data on a smaller wind turbine. The assessment revealed a good agreement between numerical and experimental results. While a weakened rotor wake was observed at the distance of 22.5 diameters downstream of the balloon turbine, the balloon wake disappeared at about 0.6 of that distance in all the wind scenarios. Vortices generated by the rotor and balloon started to merge at the tilt angle of $10°$, which intensified the turbulence intensity at 10 diameters downstream of the turbine for the wind speeds of 7 m s$^{-1}$ and 10 m s$^{-1}$. By increasing the tilt angle, the lift force on the wings experienced a sharper increase with respect to that of the whole balloon, which signified a controlling system requirement for balancing such an extra lift force.

2. The introduction is complete and extensive, but lacks some structure. In particular, two questions remain unanswered: 1) what are the advantages and disadvantages of balloon turbines with respect to other AWE systems?; 2) what is the common layout that is

expected for wind farms using this technology and how does it justify a high-fidelity  study of the wake of these machines?;

We provide more information about the merits and demerits of balloon wind turbines, the preferred wind farm layout for them, and the reasons for conducting a high-fidelity study of their wake in Line (35) of the manuscript as follows:

Line (35): Among these, the balloon wind turbine, also known as the buoyant airborne turbine (Altaeros, 2022), has relatively simpler take-off and landing maneuvers due to the buoyancy effect, which makes it suitable for deployment in various locations and temporary power generation where rapid setup is required. Moreover, these machines have minimal visual impact and can operate at higher altitudes, 400-1000 m, where wind speeds tend to be stronger and more consistent, which allows them to harness more energy compared to other AWE systems operating at lower altitudes. However, these turbines may have limitations in terms of scalability for large-scale energy generation projects. Also, employing these turbines requires adherence to aviation regulations and safety standards. And, Servicing and repairing components at high altitudes or remote locations may involve additional logistical complexities, and costs which can be complex. To mitigate these limitations, balloon wind turbines can be installed closely together within a rectangular layout. The proximity of turbines allows for satisfying higher power output requirements, efficient use of the available airspace, and enables easier maintenance and control of the turbines. In wind farms for these turbines, the wake of one turbine can affect the performance and efficiency of neighboring turbines. A high-fidelity study allows for a detailed analysis of wake interactions, including the velocity deficits, turbulence, and impact on power generation, which can help optimize the layout and spacing of the turbines.

3. The sections 2, 3.1 and 3.2 are very general and probably redundant for a technical publication like this. Please reduce them, reporting only the details specific to this work;

The governing equations of LES and sub-grid stress model for turbulence modeling were removed from section 2. General descriptions of Momentum and Balde element theories were eliminated from sections 3.1 and 3.2. Moreover, primary equations in these theories, i.e., equations 1, 2, 13, 14, and 15, and their descriptions were eliminated from the previous manuscript. Nevertheless, the secondary equations resulting from the substitution of various variables in these equations were retained to convey the fundamental principles of the BEM theory.

4. Section 3.3: the use of XFoil in the preliminary design phase is acceptable (please refer to the approach, not the airfoilTools database). However, its use for final design and simulation is questionable, especially for airfoils like the S809 with pronounced stall characteristics. At least a validation of the adopted polars with the experiments available in the literature is needed. It would also useful to specify here what conditions are gonna be simulated and how the turbine is controlled;

To choose the wind turbine in the first step, we attempted to select a rotor with available experimental data, such as NREL Phase VI. However, the dimensions of the experimentally studied rotor would not allow us to locate them inside the balloon. Therefore, we decided to customize NREL Phase VI's geometry in agreement with the balloon geometry's dimension. In this respect, the turbine's blade was divided into eight sections, and NREL Phase VI's airfoil (S809) was employed to generate the blade profile at each section. The Q-blade software was then utilized to calculate the optimized pitch angle and chord length at each section by importing the inflow details and operating conditions. To optimize the pitch angle by the program, the AOA was computed for each section using the tip speed ratio and the following equation:

$$\alpha = \tan^{-1}\left(\frac{2}{3}\frac{1}{\lambda_r}\right)$$

Optimize for lift/drag sets the twist at the specified. $\lambda_r$ at which the blade section operates to an AoA that yields the highest glide ratio. The chord distribution was optimized according to BETZ (Gasch and Twele, 2010):

$$c(r) = \frac{16\pi r}{B C_L}\sin^2(\frac{1}{3}\tan^{-1}\left(\frac{R}{r\lambda_r}\right))$$

The description of the optimization method in Qblade was added to Line (175) of the manuscript.

In our specific case, the pitch angle and chord length of the airfoils used in the blade's profile were customized specifically for their application in balloon turbines. Unfortunately, since there is no existing literature that experimentally studies the aerodynamics of this particular blade design, we were unable to validate the adopted polar against such results.

The QBlade software is a widely recognized and validated tool within the wind energy community. QBlade utilizes a blade element momentum (BEM) method and incorporates a range of empirical and theoretical models to calculate the aerodynamic performance of wind turbine blades.

By using QBlade, we were able to generate polar data for the customized airfoil profiles under the specific operating conditions of our balloon turbine design. While direct experimental validation against literature data was not feasible due to the unique nature of our blade design, the use of QBlade allowed us to capture the key characteristics of the airfoils.

5. First part of Section 4 should be moved to Section 3, in order to complete the overview of the selected test case. To what real life operating conditions do the selected tilt angles correspond?;

Thank you for your valuable suggestion. We have moved the first part of Section 4 to Section 3 to provide a complete overview of the selected test case.

The wind speeds at 400 m above the ground was estimated by using averaged meteorological data and the power law formula. The corresponding data for wind direction at the desired altitude was

unavailable, and we chose the common tilt angles for the conventional wind turbine near the ground.

6. Line 234: what criterion was used for the computation of the timestep?;

The criterion used for the computation of the time step was added to Line (215) of the manuscript as follows:

Line (215): The size of the time step was selected after sensitivity studies to assess the impact of the time step size on the convergence. The minimum time step leading to the convergence of the simulations was selected, which was approximately close to the rotor rotation of $2.5°(\Delta t = 0.0016$ s). LES calculations were run sufficiently to reach stable statistics of the flow.

7. In the majority of the results there is a typo: "separation" instead of "separation"

Thank you for pointing out the typo, and we have now corrected "seperation" to "separation" throughout the relevant sections of the paper.

---

## Author Comment (AC3)

Response to Anonymous Referee #3

Authors' responses to reviewer comments appear in blue text. Line numbers referenced in the authors' responses refer to the revised document. Figures with Arabic numerals (e.g., Figure 10) correspond to the revised manuscript; figures with Roman numerals (e.g., Figure iv) only appear in response to the reviewer's comments.

We sincerely thank the reviewer for their valuable comments, which have greatly improved our manuscript.

The present studies are intended at investigating the wake characteristics of a balloon wind turbine using an actuator disk theory solved through an LES solver. Although the topic itself is of interest, I do feel the authors need to improve the paper for publication. The following aspects are some comments from my side:

1. The authors provide too detailed formulations on the LES and actuator disk theories while they do not attempt to improve them. These can be omitted.

Thank you for your feedback. We have carefully considered your comment and made revisions to reduce the extent of this section while still ensuring clarity and comprehension. The governing equations of LES and sub-grid stress model for turbulence modeling were removed from section 2. General descriptions of Momentum and Balde element theories were eliminated from sections 3.1 and 3.2. Moreover, primary equations in these theories, i.e., equations 1, 2, 13, 14, and 15, and their descriptions were eliminated from the sections. Nevertheless, the secondary equations resulting from the substitution of various variables in these equations were retained to convey the fundamental principles of the BEM theory.

2. Numerical description is too weak. The authors shall provide more information about the approaches used to solve the sets of flow equations, e.g., time integration? discretization? convergence acceleration?

We added more information about the simulation set up in Line (210) of the manuscript as follows:

Line (210): The PISO scheme was utilized for pressure-velocity coupling. This scheme is well-suited for unsteady and highly transient flows, which are characteristic of wind turbine wake simulations. For spatial discretization of pressure and momentum, the second-order form and the time integration, the second-order implicit, were employed to improve the stability and convergence of the simulations. Simulations ran for a maximum of 20 iterations per time step, using a convergence criterion of $1 \times 10^{-4}$ for the residuals in all cases.

3. Temporal discretization studies are not performed. How can we sure the solutions are accurate with respect to time size?

Due to the inherent complexity and computational demands of LES, conducting a comprehensive investigation into the accuracy with respect to time size would have imposed significant limitations on our computational resources. However, we would like to emphasize that we took steps to ensure the accuracy and convergence of our simulations in terms of the time step size. Sensitivity studies were conducted to assess the impact of the time step size on the convergence of the simulations. After careful consideration, we selected a minimum time step that led to the convergence of the simulations, which was approximately close to the rotor rotation of $2.5°(\Delta t = 0.0016 \text{ s})$.

The criterion used for the computation of the time step was added to Line (215) of the manuscript as follows:

Line (215): The size of the time step was selected after sensitivity studies to assess the impact of the time step size on the convergence. The minimum time step leading to the convergence of the simulations was selected, which was approximately close to the rotor rotation of $2.5°(\Delta t = 0.0016 \text{ s})$. LES calculations were run sufficiently to reach stable statistics of the flow.

4. Upstream domain looks a bit too small.

To complete the determination of the domain dimensions, it was solved separately by different domain sizes and observing the variable flow gradients at the boundaries. The computational domain was chosen as the minimum size that exhibited zero gradients at the boundaries. Figure i illustrates the pressure gradients for $U_{ref} = 7 \text{ m s}^{-1}$ and $\theta_{tilt} = 0°, 5°$ and $10°$ on a symmetry plane of the balloon in the finalized domain. The pressure contours provide an evident indication that there is no pressure gradient present at the boundary of the domain with the determined dimensions.

[Figure]

(a)

(b)

[Figure]

Pressure.Gradient
Contour 1

6.0e+01
5.6e+01
5.3e+01
5.0e+01
4.6e+01
4.3e+01
4.0e+01
3.6e+01
3.3e+01
3.0e+01
2.7e+01
2.3e+01
2.0e+01
1.7e+01
1.3e+01
9.9e+00
6.6e+00
3.3e+00
0.0e+00
[kg m^-2 s^-2]

(c)

Figure i. Pressure gradient contour for $U_{ref}$ = 7 m s$^{-1}$ and(a) $\theta_{tilt} = 0°$, (b) $\theta_{tilt} = 5°$ (c) $\theta_{tilt} = 10°$ on the symmetry plane of the balloon (z=0).

5. Validation studies use conventional wind turbines, while the case being studied is much more complex.

Due to the novelty of balloon wind turbines, there is a lack of experimental data in the literature to validate the numerical results pertaining to their aerodynamics and wake flow. Nevertheless, it is possible to assess the methodology employed in this study to characterize the wake flow of these turbines. To this end, we employed the same methodology to investigate the wake behavior of a smaller turbine that had been previously studied experimentally.

There are two key differences between the balloon wind turbine and the smaller turbine: the diameter of their rotors and the presence of the balloon in the main model. Based on the results obtained from the balloon wind turbine, it is evident that the effective length of the rotor wake is considerably longer than that of the balloon, and they do not interact under various inflow conditions considered in this study. Hence, the wake flow behavior of the rotor is independent of the balloon.

Although the turbine being studied differs in diameter from the smaller turbine used for validation, it is still significant to validate our methodology using a smaller turbine with available experimental data. The purpose of this validation was to demonstrate the accuracy and reliability of the numerical approach, irrespective of the specific turbine size or complexity.

Furthermore, the utilization of LES enables high-resolution analysis of the turbulent flow structures within the turbine wake. By adequately resolving the flow features, our model captures the intricate complexities associated with different turbine sizes. The results obtained from the validation study showcased a satisfactory agreement between the LES-ADM method and experimental measurements. Therefore, in the absence of experimental data for the balloon wind turbine, we argue that validating the robustness of our methodology through promising outcomes obtained using the same approach serves as a reliable indicator of the accuracy of the main model's findings.

6.  If the authors claim to use LES, is the energy in the proximity of the balloon wall resolved well?

In LES, the $y^+$ values should be lower than 5 for wall-resolved LES simulations. Moreover, the Dynamic Smagorinsky model tends to perform better in capturing near-wall turbulence compared to traditional static SGS models, as it adapts the SGS model coefficients dynamically based on the local flow characteristics. When using this model, a lower y Plus value (closer to 1) is often recommended for accurate resolution of the near-wall turbulent structures. This is because the model is designed to capture small-scale motions more effectively, and a finer resolution near the wall helps to capture the important energy-containing turbulent eddies. When the $y^+$ value falls within this range, it generally indicates that the energy near the wall is resolved well. To reach these values near the balloon wall in the simulations, the mesh size and its spacing near the wall were adjusted accordingly. Figure ii illustrates $y^+$ contours for $U_{ref} = 7$ m s$^{-1}$ and $\theta_{tilt} = 0°, 5°$ and $10°$ on the balloon wall. According to the figure, the value of this parameter is close to 1 in most regions on the wall and does not exceed 3.5 in any region for different wind scenarios.

[Figure]

(a)

[Figure]

(b)

(c)

Figure ii. Wall Yplus contours for $U_{ref} = 7$ m s$^{-1}$ and (a) $\theta_{tilt} = 0°$, (b) $\theta_{tilt} = 5°$ (c) $\theta_{tilt} = 10°$ on the balloon's wall.

7. The authors focus the studies on the wake, but the mesh behind the balloon does not seem to be well refined such that it can well resolve small eddies. Perhaps plot the Q or Lambda2 criterion?

To determine the mesh distribution in the wake region, LES mesh criterion, which is presented in section 5.1, was adopted. The mesh size in the near-wake is illustrated in Figure iii (a). To resolve smaller eddies behind the turbine and balloon's separation region, the mesh sizing in these areas were refined to be smaller, which is evident in the figure. The magnitude of Lambda 2 is clipped for smaller eddies, and the corresponding contour is depicted in Figure iii (b). It is evident that the small eddies in the separation zone of the balloon and the wake of the turbine are properly resolved.

[Figure]

(a)

[Figure]

(b)

Figure iii. (a) Mesh distribution (b) Lambda 2 contour on the symmetry plane of the balloon

8. Last minor aspect: remove the 6th point in Conclusion as nothing is written there.

Thank you for pointing out this error; we have removed it.

---

## Author Comment (AC4)

Response to Anonymous Referee #1

Authors' responses to reviewer comments after major revision appear in purple text.

We would like to express our sincere gratitude to the referee for their valuable time and effort in reviewing our paper, as well as for their insightful suggestions.

Authors' responses to reviewer comments appear in blue text. Line numbers referenced in the authors' responses refer to the revised document. Figures with Arabic numerals (e.g., Figure 5) correspond to the revised manuscript; figures with Roman numerals (e.g., Figure vi) only appear in response to the reviewer's comments.

1. Little bit of background is required in the abstract;

Thank you for your comment regarding the background information in the abstract. In the updated abstract, we have provided more details about airborne wind energy systems and particularly balloon wind turbines, as below:

In the realm of novel technologies for generating electricity from renewable resources, an emerging category of wind energy converters called Airborne Wind Energy Systems (AWESs) has gained prominence. These pioneering systems employ tethered wings or aircraft that operate at higher atmospheric layers, enabling them to harness wind speeds surpassing conventional wind turbines' capabilities. The balloon wind turbine is one type of AWESs that utilizes the buoyancy effect to elevate the turbine to altitudes typically ranging from 400 to 1000 meters. In this paper, the wake characteristics and aerodynamics of a balloon wind turbine were numerically investigated for different wind scenarios. Large eddy simulation, along with the actuator disk model, was employed to predict the wake behavior of the turbine. To improve the accuracy of the simulation results, a structured grid was generated and refined by using an algorithm to resolve about 80% of the local turbulent kinetic energy in the wake. Results contributed to designing an optimized layout of wind farms and stability analysis of such systems. The capabilities of the hybrid ADM-LES model when using the mesh generation algorithm were evaluated against the experimental data on a smaller wind turbine. The assessment revealed a good agreement between numerical and experimental results. While a weakened rotor wake was observed at the distance of 22.5 diameters downstream of the balloon turbine, the balloon wake disappeared at about 0.6 of that distance in all the wind scenarios. Vortices generated by the rotor and balloon started to merge at the tilt angle of $10°$, which intensified the turbulence intensity at 10 diameters downstream of the turbine for the wind speeds of 7 m s$^{-1}$ and 10 m s$^{-1}$. By increasing the tilt angle, the lift force on the wings experienced a sharper increase with respect to that of the whole balloon, which signified a controlling system requirement for balancing such an extra lift force.

2. The introduction is complete and extensive, but lacks some structure. In particular, two questions remain unanswered: 1) what are the advantages and disadvantages of balloon turbines with respect to other AWE systems?; 2) what is the common layout that is expected for wind farms using this technology and how does it justify a high-fidelity study of the wake of these machines?;

We provide more information about the merits and demerits of balloon wind turbines, the preferred wind farm layout for them, and the reasons for conducting a high-fidelity study of their wake in Line (35) of the manuscript as follows:

Line (35): Among these, the balloon wind turbine, also known as the buoyant airborne turbine (Altaeros, 2022), has relatively simpler take-off and landing maneuvers due to the buoyancy effect, which makes it suitable for deployment in various locations and temporary power generation where rapid setup is required. Moreover, these machines have minimal visual impact and can operate at higher altitudes, 400-1000 m, where wind speeds tend to be stronger and more consistent, which allows them to harness more energy compared to other AWE systems operating at lower altitudes. However, these turbines may have limitations in terms of scalability for large-scale energy generation projects. Also, employing these turbines requires adherence to aviation regulations and safety standards. And, Servicing and repairing components at high altitudes or remote locations may involve additional logistical complexities, and costs which can be complex. To mitigate these limitations, balloon wind turbines can be installed closely together within a rectangular layout. The proximity of turbines allows for satisfying higher power output requirements, efficient use of the available airspace, and enables easier maintenance and control of the turbines. In wind farms for these turbines, the wake of one turbine can affect the performance and efficiency of neighboring turbines. A high-fidelity study allows for a detailed analysis of wake interactions, including the velocity deficits, turbulence, and impact on power generation, which can help optimize the layout and spacing of the turbines.

3. The sections 2, 3.1 and 3.2 are very general and probably redundant for a technical publication like this. Please reduce them, reporting only the details specific to this work;

The governing equations of LES and sub-grid stress model for turbulence modeling were removed from section 2. General descriptions of Momentum and Bladed element theories were eliminated from sections 3.1 and 3.2. Moreover, primary equations in these theories, i.e., equations 1, 2, 13, 14, and 15, and their descriptions were eliminated from the previous manuscript. Nevertheless, the secondary equations resulting from the substitution of various variables in these equations were retained to convey the fundamental principles of the BEM theory.

4. Section 3.3: the use of XFoil in the preliminary design phase is acceptable (please refer to the approach, not the airfoilTools database). However, its use for final design and simulation is questionable, especially for airfoils like the S809 with pronounced stall characteristics. At least a validation of the adopted polars with the experiments available in the literature is needed. It would also useful to specify here what conditions are gonna be simulated and how the turbine is controlled;

To choose the wind turbine in the first step, we attempted to select a rotor with available experimental data, such as NREL Phase VI. However, the dimensions of the experimentally studied rotor would not allow us to locate them inside the balloon. Therefore, we decided to

customize NREL Phase VI's geometry in agreement with the balloon geometry's dimension. In this respect, the turbine's blade was divided into eight sections, and NREL Phase VI's airfoil (S809) was employed to generate the blade profile at each section. The Q-blade software was then utilized to calculate the optimized pitch angle and chord length at each section by importing the inflow details and operating conditions. To optimize the pitch angle by the program, the AOA was computed for each section using the tip speed ratio and the following equation:

$$\alpha = \tan^{-1}\left(\frac{2}{3}\frac{1}{\lambda_r}\right)$$

Optimize for lift/drag sets the twist at the specified. $\lambda_r$ at which the blade section operates to an AoA that yields the highest glide ratio. The chord distribution was optimized according to BETZ (Gasch and Twele, 2010):

$$c(r) = \frac{16\pi r}{BC_L}sin^2(\frac{1}{3}tan^{-1}\left(\frac{R}{r\lambda_r}\right))$$

The description of the optimization method in Qblade was added to Line (175) of the manuscript.

In our specific case, the pitch angle and chord length of the airfoils used in the blade's profile were customized specifically for their application in balloon turbines. Unfortunately, since there is no existing literature that experimentally studies the aerodynamics of this particular blade design, we were unable to validate the adopted polar against such results.

The QBlade software is a widely recognized and validated tool within the wind energy community. QBlade utilizes a blade element momentum (BEM) method and incorporates a range of empirical and theoretical models to calculate the aerodynamic performance of wind turbine blades.

By using QBlade, we were able to generate polar data for the customized airfoil profiles under the specific operating conditions of our balloon turbine design. While direct experimental validation against literature data was not feasible due to the unique nature of our blade design, the use of QBlade allowed us to capture the key characteristics of the airfoils.

5. First part of Section 4 should be moved to Section 3, in order to complete the overview of the selected test case. To what real life operating conditions do the selected tilt angles correspond?;

Thank you for your valuable suggestion. We have moved the first part of Section 4 to Section 3 to provide a complete overview of the selected test case.

The wind speeds at 400 m above the ground was estimated by using averaged meteorological data and the power law formula. The corresponding data for wind direction at the desired altitude was unavailable, and we chose the common tilt angles for the conventional wind turbine near the ground.

6. Line 234: what criterion was used for the computation of the timestep?;

The criterion used for the computation of the time step was added to Line (215) of the manuscript as follows:

Line (215): The size of the time step was selected after sensitivity studies to assess the impact of the time step size on the convergence. To avoid excessive computational costs, the maximum time step required for the simulations to converge was selected, which was approximately close to the time required for a rotor rotation of 2.5 degrees ($\Delta t = 0.0016$ s). LES calculations were run sufficiently to reach stable statistics of the flow.

In response to your query regarding the criterion employed for determining the time step, we wish to provide a comprehensive view of our approach. Initially, we considered multiple methods for selecting the time step size, including an evaluation of the independence of results from the time step and an assessment against the minimum flow time scale in critical regions. However, after careful deliberation and consideration of our computational resources, we opted to primarily rely on the convergence criterion. This decision was driven by the merits of the convergence criterion, which had a robust track record and was meticulously established through sensitivity studies to ensure both the stability and accuracy of our simulations. Nonetheless, we present the results of our additional methods here to underscore the robustness and reliability of our approach.

**1. Independence of the results from the selected time step:**

To address this, we duplicated a similar simulation setup as detailed in Section 4 of our paper, with $U_{ref}$ set to 7 and $\theta_{tilt}$ at $0°$. We conducted simulations employing smaller time steps (specifically, 0.0004 s and 0.008 s) compared to the time step used in the original simulations. It is important to note that when using larger time steps, we encountered numerical instability that failed to meet the convergence criterion. Figure i illustrates the vertical profiles of the time-averaged normalized x-velocity at various positions downstream of the wind turbine, spanning the range of -4 < y/d < 4 at z = 0.

[Figure]

(a)                              (b)                              (c)

**Figure i. Comparison of vertical profiles of the time-averaged normalized x-velocity for different time steps for $U_{ref} = 7$ m s$^{-1}$ and -4<y/d<4, and z=0 with $\theta_{tilt} = 0°$ at (a) x/d= 3 (b) x/d= 7 (c) x/d= 14.**

According to figure i, the difference in the time-averaged normalized x-velocity at x/d = 3 for simulations using $\Delta t = 0.0016$ s and $\Delta t = 0.004$ s is found to be below 1 percent. Moreover, this difference further diminishes in downstream locations. The negligible variation observed between the original time step and the smaller time steps demonstrates that the chosen $\Delta t$ is sufficiently

small to accurately capture the intricate details of the flow. This includes the representation of small-scale turbulent features and the unsteady behaviour of the wake, which are directly linked to the precise prediction of velocity deficits in the wake region.

**2. Time scale method:**

The selected time step was then assessed to ensure it remained lower than or close to the smallest time scale governing the flow dynamics within our critical areas of interest —specifically, in the wake zone and the vicinity of the balloon. In this refined approach, we considered cells within these regions with the largest edge length ($\Delta x$) and local average velocity (V). We aimed to ensure that our time step ($\Delta t$) was sufficiently small to provide the necessary temporal resolution to accurately capture the flow's behaviour as it traversed these cells ($\Delta t \leq \Delta x / V$). In this regard, LES calculations were run sufficiently to reach stable statistics of the flow. The time scale was then computed using Eq. (1). Time scale contours in the iso-clipped symmetry plan of the balloon ($z=0$ and $-9<y<9$) for $U_{ref} = 7$, 10 m s$^{-1}$ and $\theta_{tilt} = 0°$ are shown in figure ii.

$$Time\ scale = \frac{cell\ volume^{\frac{1}{3}}}{|V|}$$ (1)

[Figure]

(a)

(b)

**Figure ii. Time scale contours in the iso-clipped symmetry plan of the balloon (z=0 and -9<y<9) for $\theta_{tilt} = 0°$**
**(a) $U_{ref} = 7$ m s$^{-1}$ and (b) $U_{ref} = 10$ m s$^{-1}$**

As depicted in figure ii, the minimum time scale within the wake region and surrounding the balloon remains within the range of 0.0025 seconds for $U_{ref} = 7$ m s$^{-1}$ and 0.0018 seconds for $U_{ref} = 10$ m s$^{-1}$, which falls comfortably below the time step determined through sensitivity studies for convergence (0.0016 seconds). Further examination of the time scale and independence of the results from the selected time step under varying conditions, specifically for $\theta_{tilt} = 5°$ and $10°$ at both inlet reference velocities, serves to confirm that the chosen time step aligns with both aforementioned criteria.

7. In the majority of the results there is a typo: "separation" instead of "separation"

Thank you for pointing out the typo, and we have now corrected "seperation" to "separation" throughout the relevant sections of the paper.

---

## Author Comment (AC5)

Response to Referee #2: Joshua Brinkerhoff

Authors' responses to reviewer comments after major revision appear in purple text.

We would like to express our sincere gratitude to the referee for their valuable time and effort in reviewing our paper, as well as for their insightful suggestions.

Authors' responses to reviewer comments appear in blue text. Line numbers referenced in the authors' responses refer to the revised document. Figures with Arabic numerals (e.g., Figure 10) correspond to the revised manuscript; figures with Roman numerals (e.g., Figure iv) only appear in response to the reviewer's comments.

1. Overall, I found the paper to be rather lengthy in its description of the AD method and associated momentum theory, which is not novel.

Thank you for your feedback on the length of the paper's description regarding the AD method and associated momentum theory. We have carefully considered your comment and made revisions to reduce the extent of this section while still ensuring clarity and comprehension. In this respect, general descriptions of Momentum and Blade element theories were eliminated from sections 3.1 and 3.2. Moreover, primary equations in these theories, i.e., equations 1, 2, 13, 14, and 15, and their descriptions were removed from the sections. Nevertheless, the secondary equations resulting from the substitution of various variables in these equations were retained to convey the fundamental principles of the BEM theory.

2. I expected to see the sensitivity of the computational domain size, which was not provided and my intuition suggests is small, especially the upstream distance between the inlet and the balloon turbine.

To complete the determination of the domain dimensions, it was solved separately by different domain sizes and observing the variable flow gradients at the boundaries. The computational domain was chosen as the minimum size that exhibited zero gradients at the boundaries. Figure i illustrates the pressure gradients for $U_{ref} = 7$ m s$^{-1}$ and $\theta_{tilt} = 0°, 5°$ and $10°$ on a symmetry plane of the balloon in the finalized domain. The pressure contours provide an evident indication that there is no pressure gradient present at the boundary of the domain with the determined dimensions.

[Figure]

**(a)**

**(b)**

[Figure]

**Figure i. Pressure gradient contour for U$_{ref}$ = 7 m s$^{-1}$ and(a) $\theta_{tilt} = 0°$, (b) $\theta_{tilt} = 5°$ (c) $\theta_{tilt} = 10°$on the symmetry plane of the balloon (z=0).**

In response to the referee's query regarding the sensitivity of the computational domain size, we would like to provide additional information to further support the appropriateness of the domain size chosen in our study. In our initial response, we explained that the domain dimensions were carefully determined by evaluating flow gradients at the boundaries, to select a size that ensured zero gradients. While this method was valid, we employed sensitivity analysis of wake characteristics in relation to domain size here to further support the reliability of our approach. In light of this, we have conducted a follow-up study wherein we created two different computational domains: one with an extended upstream length (from 5d to 10d) and another with an extended downstream length (from 22.5d to 30d) as shown in figure ii relative to the turbine position, beyond the original domain utilized in the research. We increased the number of nodes in the upstream distance by a factor of 2 for the extended upstream domain and by 1.3 times (equal to the length increment ratio) for the extended downstream domain to ensure consistent spatial resolution in all cases, while only considering changes in the computational domain size on the results.

[Figure]

**Figure ii. Computational domain size for (a) extended upstream length and (b) extended downstream length.**

With these new computational domains, we applied the same simulation settings and boundary conditions as described in Section 4 of the paper to conduct simulations for the cases where $U_{ref}$ =7 m s$^{-1}$ and $\theta_{tilt} = 0°$ and $10°$. Figure iii showcases the simulation results employing an extended upstream length, illustrating vertical profiles of the time-averaged normalized x-velocity at x/d = -2, within the range of -4 < y/d < 4 at z = 0. Additionally, figure iv presents the outcomes of the simulation utilizing an extended downstream length, displaying vertical profiles of the time-averaged normalized x-velocity at x/d = 5 and 14, within the range of -4 < y/d < 4 at z = 0.

[Figure]

**Figure iii. Comparison of vertical profiles of the time-averaged normalized x-velocity for different upstream lengths for $U_{ref} = 7$ m s$^{-1}$ and -4<y/d<4, and z=0 with $\theta_{tilt}=$ (a) $0°$ (b) $10°$ at x/d =-2.**

[Figure]

**Figure iv. Comparison of vertical profiles of the time-averaged normalized x-velocity for different downstream lengths for $U_{ref} = 7$ m s$^{-1}$ and -4<y/d<4, and z=0 with $\theta_{tilt}= 0°$ at (a) x/d =5 (b) x/d= 14 and $\theta_{tilt}= 10°$ at (c) x/d =5 (d) x/d= 14.**

According to figure iii, extending upstream lengths has been observed to have a relatively minor impact on the velocity profiles at various locations relative to the wind turbine position. This observation can be ascribed to the specific conditions governing our investigation. Notably, we considered our wind turbine situated at a high altitude where the atmosphere tends to be more stable because it is less influenced by surface heating and friction, which can lead to reduced turbulence and vertical mixing. The absence of significant boundary layer effects due to the high-altitude location of our wind turbine led to a longer upstream length less critical for capturing boundary layer-related phenomena. These specific environmental conditions enabled us to design our computational domain with a smaller upstream length than typically required for studying ground-based wind turbine wake behaviour, prioritizing computational efficiency while maintaining result accuracy.

Furthermore, as shown in figure iv, prolonging the downstream distances has shown only a marginal influence on the velocity profiles at different positions relative to the wind turbine's location. Our choice of downstream length was carefully considered in light of several critical factors. The selected downstream domain size was designed to encompass the essential characteristics of the wake, including wake recovery, turbulence decay, and gradual mixing with ambient air. This careful consideration of downstream length was paramount to accurately capturing the wake's behaviour and its impact on downstream flow. In summary, under these controlled conditions and with careful attention to factors critical to wake simulation, the impact of extending the domain size upstream and downstream remained minimal, providing robust support for the appropriateness of our chosen computational domain size.

3. Secondly, the analysis to ensure consistent spatial resolution relies on a RANS simulation for estimating the turbulence kinetic energy and dissipation rate for calculating the turbulence integral scale. The details of the RANS simulation are not provided. Moreover, why the RANS solution can be considered accurate is not justified.

The details of the RANS simulations and the reasons for choosing this model to calculate turbulence kinetic energy and dissipation rate are added in Line (257) of the manuscript as follows:

Line (257): The K-Omega SST model was employed in the precursor simulations, utilizing simulation setup and boundary conditions similar to those described in section 4 of the paper, duplicating the main model configuration. The K-Omega SST model accurately estimates turbulence kinetic energy and dissipation by employing a dual-equation formulation, which captures the interactions between these quantities more comprehensively. Additionally, its enhanced near-wall treatment improves accuracy in capturing boundary layer characteristics around the balloon surface, making it a reliable choice for precise calculations.

In response to the referee's inquiry regarding the accuracy and reliability of the RANS solutions used in our methodology for generating LES-friendly meshes, we appreciate the opportunity to provide a more detailed explanation and justification.

**Verification and Validation:** To ensure the accuracy of the RANS solutions leading to the generation of LES-friendly meshes, we undertook several measures:

1. **Comparison with Experimental Data:** In the case of smaller wind turbine where experimental data were available, our LES-friendly mesh generation approach was applied to simulate the wake behaviour of the turbine. The results derived from this approach exhibited a good level of agreement with the corresponding experimental data, providing persuasive evidence for the appropriateness of the mesh generated based on the RANS simulation results.
2. **Grid Independence:** We rigorously assessed the quality of the mesh generation algorithm by conducting grid independence studies for both the smaller turbine and balloon wind turbine simulations as described in response to questions 4 and 5 in this document. Our methodology consistently satisfied grid independence criteria, indicating the mesh's suitability and the reliability of the RANS solutions.
3. **Convergence Criterion:** Our simulations consistently met the convergence criterion of $1 \times 10^{-4}$ for residuals across all cases. This demonstrates the stability and convergence of the RANS solutions, further affirming their accuracy.

In cases where experimental data were not available for the balloon wind turbine study, we acknowledge the limitation of direct experimental validation. However, we firmly believe that the combined evidence from the successful agreement with experimental data in a similar case, grid independence, and convergence criteria support the appropriateness and accuracy of the RANS solutions employed for LES-friendly mesh generation.

4. Thirdly, the grid independence assesses the pressure coefficient distribution along the balloon periphery, which is not convincing for assessing grid independence of the results. More convincing would be to demonstrate the grid independence of the wake recovery, separation zone size and strength, and other parameters that would be expected to be more sensitive to the grid.

Thank you for your valuable comment regarding the assessment of grid independence in our paper. We appreciate your suggestion and would like to address your concern. In our previous evaluation of grid independence, we focused on the pressure coefficient distribution along the balloon periphery. While this aspect provides some insights into the grid independence of aerodynamic loads on balloons, we acknowledge that it may not be the most convincing parameter for evaluating grid independence of the wake characteristics, which is the leading study concern. Therefore, to avoid excessive length in this section resulting from the inclusion of grid independence analysis for both parameters, we have opted to focus our analysis on the grid size effect, specifically on wake recovery. Consequently, we have made the necessary revisions in Line (272) in the manuscript as follows:

Line (272): To further assess the criterion, its grid independence was investigated. Accordingly, two coarser (G1) and finer (G3) meshes, summarized in Table 1, were generated. By employing these meshes in three simulations, a comparison was conducted on the vertical profiles of the time-averaged normalized x-velocity at three distinct downstream locations. All of the simulations were performed for $U_{\text{ref}} = 10$ m s$^{-1}$ and $\theta_{tilt} = 0°$ and the results are illustrated in Figure 10. According to Figure 10, using a coarser grid in the near wake leads to a lower prediction of velocity deficit. This is because as the grid size grows, the small-scale turbulence structures and vortices are not

accurately resolved. As a result, the flow tends to smooth out, and the turbulence effects are underestimated. This can lead to an underprediction of the velocity deficit in the near wake. However, the difference in average velocity at 3d downstream of the turbine between mesh G3 and G2 is about 1%, while this difference is around 4% for meshes G2 and G1. Moreover, the difference between velocity profiles for different grids decreases in further regions. Since the discrepancy between G2 and G1 mesh results is about one-fourth of the difference between G2 and G3, the mesh criterion in LES satisfies the wake recovery's mesh independence requirement.

**Table 1**
**Mesh distribution in the computational domain for evaluating mesh criterion in LES.**

| Grid number | G1 | G2 | G3 |
|---|---|---|---|
| Nodes on edge Nin | 23 | 43 | 50 |
| Nodes on edge Nwing | 25 | 35 | 40 |
| Nodes on edge Nout | 25 | 30 | 40 |
| Nodes on edge Nu | 20 | 30 | 40 |
| Nodes on edge Ns | 74 | 86 | 95 |
| Nodes on edge Nd | 250 | 285 | 300 |
| Nodes on edge Np | 140 | 196 | 236 |
| Nodes within the boundary layers Nbl | 35 | 35 | 35 |
| Total Number of nodes Nt | 4,972,096 | 10,756,364 | 15,4657,804 |

[Figure]

**Figure 10. Vertical profiles of the time-averaged x-velocity at different locations downstream of the turbine.**

5.   Fourthly, the validation against experiment is not well documented--the authors do not comment on the spatial resolution of the validation study and whether it is consistent with the main study. The result is that the validation--which does show good agreement--does not convincingly demonstrate the accuracy of the main study results.

To clarify the spatial resolution of the validation study, we add the node distribution in the domain in Line (301) of the manuscript as follows:

Line (301): The cubic domain was discretized with 192, 32, and 42 nodes along the x and y axes.

Moreover, we added a section in Line (319) of the manuscript to evaluate the grid independence of the results of the validation study as below:

Line (319): To assess the grid independence of velocity profiles in the wake, the simulations were performed for a coarser and finer mesh. The number of nodes in the coarser and finer grids along

the x, y, and z directions was 120×25×30 and 250×50×60 respectively. Figure 14 shows the comparison of vertical profiles of the time-averaged streamwise velocity obtained from the experimental study and 3 different grids. According to Figure 14, decreasing the grid size has a minor effect on the velocity profiles within the wake, and the results obtained from the main grid demonstrate good consistency with experimental measurements.

[Figure]

**Figure 14. Comparison of vertical profiles of the time-averaged streamwise velocity U (m s$^{-1}$) obtained by the experiment and different grids.**

---

## Author Comment (AC6)

Response to Anonymous Referee #3

Authors' responses to reviewer comments after major revision appear in purple text.

We would like to express our sincere gratitude to the referee for their valuable time and effort in reviewing our paper, as well as for their insightful suggestions.

Authors' responses to reviewer comments appear in blue text. Line numbers referenced in the authors' responses refer to the revised document. Figures with Arabic numerals (e.g., Figure 10) correspond to the revised manuscript; figures with Roman numerals (e.g., Figure iv) only appear in response to the reviewer's comments.

The present studies are intended at investigating the wake characteristics of a balloon wind turbine using an actuator disk theory solved through an LES solver. Although the topic itself is of interest, I do feel the authors need to improve the paper for publication. The following aspects are some comments from my side:

1. The authors provide too detailed formulations on the LES and actuator disk theories while they do not attempt to improve them. These can be omitted.

Thank you for your feedback. We have carefully considered your comment and made revisions to reduce the extent of this section while still ensuring clarity and comprehension. The governing equations of LES and sub-grid stress model for turbulence modeling were removed from section 2. General descriptions of Momentum and Balde element theories were eliminated from sections 3.1 and 3.2. Moreover, primary equations in these theories, i.e., equations 1, 2, 13, 14, and 15, and their descriptions were eliminated from the sections. Nevertheless, the secondary equations resulting from the substitution of various variables in these equations were retained to convey the fundamental principles of the BEM theory.

2. Numerical description is too weak. The authors shall provide more information about the approaches used to solve the sets of flow equations, e.g., time integration? discretization? convergence acceleration?

We added more information about the simulation set up in Line (210) of the manuscript as follows:

Line (210): The PISO scheme was utilized for pressure-velocity coupling. This scheme is well-suited for unsteady and highly transient flows, which are characteristic of wind turbine wake simulations. For spatial discretization of pressure and momentum, the second-order form and the time integration, the second-order implicit, were employed to improve the stability and convergence of the simulations. Simulations ran for a maximum of 20 iterations per time step, using a convergence criterion of $1 \times 10^{-4}$ for the residuals in all cases.

To further elaborate on the previous details of the simulation setup, we decided to provide more information about the convergence acceleration method used in our research. Following is the updated description of numerical settings in the manuscript:

Line (210): The PISO scheme was utilized for pressure-velocity coupling. This scheme is well-suited for unsteady and highly transient flows, which are characteristic of wind turbine wake simulations. For spatial discretization of pressure and momentum, the second-order form and the time integration, the second-order implicit, were employed to improve the stability and convergence of the simulations. Simulations ran for a maximum of 20 iterations per time step, using a convergence criterion of $1 \times 10^{-4}$ for the residuals in all cases. Obtaining convergence in unsteady wind turbine wake simulations is time-consuming. To expedite the convergence process while maintaining stability, under-relaxation techniques, including the application of factors of 0.3 to the pressure equation and 0.7 to the momentum equation, were employed.

3. Temporal discretization studies are not performed. How can we sure the solutions are accurate with respect to time size?

Due to the inherent complexity and computational demands of LES, conducting a comprehensive investigation into the accuracy with respect to time size would have imposed significant limitations on our computational resources. However, we would like to emphasize that we took steps to ensure the accuracy and convergence of our simulations in terms of the time step size. Sensitivity studies were conducted to assess the impact of the time step size on the convergence of the simulations. After careful consideration, we selected a maximum time step required for the simulations to converge, which was approximately close to the time required for a rotor rotation of 2.5 degrees ($\Delta t = 0.0016$ s).

The criterion used for the computation of the time step was added to Line (215) of the manuscript as follows:

Line (215): The size of the time step was selected after sensitivity studies to assess the impact of the time step size on the convergence. To avoid excessive computational costs, the maximum time step required for the simulations to converge was selected, which was approximately close to the time required for a rotor rotation of 2.5 degrees ($\Delta t = 0.0016$ s). LES calculations were run sufficiently to reach stable statistics of the flow.

In response to your query regarding the criterion employed for determining the time step, we wish to provide a comprehensive view of our approach. Initially, we considered multiple methods for selecting the time step size, including an evaluation of the independence of results from the time step and an assessment against the minimum flow time scale in critical regions. However, after careful deliberation and consideration of our computational resources, we opted to primarily rely on the convergence criterion. This decision was driven by the merits of the convergence criterion, which had a robust track record and was meticulously established through sensitivity studies to ensure both the stability and accuracy of our simulations. Nonetheless, we present the results of our additional methods here to underscore the robustness and reliability of our approach.

**1. Independence of the results from the selected time step:**

To address this, we duplicated a similar simulation setup as detailed in Section 4 of our paper, with $U_{ref}$ set to 7 and $\theta_{tilt}$ at $0°$. We conducted simulations employing smaller time steps (specifically, 0.0004 s and 0.008 s) compared to the time step used in the original simulations. It is important to note that when using larger time steps, we encountered numerical instability that failed to meet the convergence criterion. Figure i illustrates the vertical profiles of the time-averaged normalized x-velocity at various positions downstream of the wind turbine, spanning the range of -4 < y/d < 4 at z = 0.

[Figure]

(a)                                    (b)                                    (c)

**Figure i. Comparison of vertical profiles of the time-averaged normalized x-velocity for different time steps for $U_{ref}$ =7 m s$^{-1}$ and -4<y/d<4, and z=0 with $\theta_{tilt}$= $0°$ at (a) x/d= 3 (b) x/d= 7 (c) x/d= 14**

According to figure i, the difference in the time-averaged normalized x-velocity at x/d = 3 for simulations using Δt = 0.0016 s and Δt = 0.004 s is found to be below 1 percent. Moreover, this difference further diminishes in downstream locations. The negligible variation observed between the original time step and the smaller time steps demonstrates that the chosen Δt is sufficiently small to accurately capture the intricate details of the flow. This includes the representation of small-scale turbulent features and the unsteady behaviour of the wake, which are directly linked to the precise prediction of velocity deficits in the wake region.

**2. Time scale method:**

The selected time step was then assessed to ensure it remained lower than or close to the smallest time scale governing the flow dynamics within our critical areas of interest —specifically, in the wake zone and the vicinity of the balloon. In this refined approach, we considered cells within these regions with the largest edge length (Δx) and local average velocity (V). We aimed to ensure that our time step (Δt) was sufficiently small to provide the necessary temporal resolution to accurately capture the flow's behaviour as it traversed these cells (Δt ≤ Δx / V). In this regard, LES calculations were run sufficiently to reach stable statistics of the flow. The time scale was then computed using Eq. (1). Time scale contours in the iso-clipped symmetry plan of the balloon (z=0 and -9<y<9) for $U_{ref}$ = 7, 10 m s$^{-1}$ and $\theta_{tilt}$ = $0°$ are shown in figure ii.

$$Time\ scale = \frac{cell\ volume^{\frac{1}{3}}}{|V|}$$

(1)

[Figure]

**Figure ii. Time scale contours in the iso-clipped symmetry plan of the balloon (z=0 and -9<y<9) for $\theta_{tilt} = 0°$ (a) $U_{ref} = 7$ m s$^{-1}$ and (b) $U_{ref} = 10$ m s$^{-1}$**

As depicted in figure ii, the minimum time scale within the wake region and surrounding the balloon remains within the range of 0.0025 seconds for $U_{ref} = 7$ m s$^{-1}$ and 0.0018 seconds for $U_{ref} = 10$ m s$^{-1}$, which falls comfortably below the time step determined through sensitivity studies for convergence (0.0016 seconds). Further examination of the time scale and independence of the results from the selected time step under varying conditions, specifically for $\theta_{tilt} = 5°$ and $10°$ at both inlet reference velocities, serves to confirm that the chosen time step aligns with both aforementioned criteria.

4. Upstream domain looks a bit too small.

To complete the determination of the domain dimensions, it was solved separately by different domain sizes and observing the variable flow gradients at the boundaries. The computational domain was chosen as the minimum size that exhibited zero gradients at the boundaries. Figure iii illustrates the pressure gradients for $U_{ref} = 7$ m s$^{-1}$ and $\theta_{tilt} = 0°, 5°$ and $10°$ on a symmetry plane of the balloon in the finalized domain. The pressure contours provide an evident indication that

there is no pressure gradient present at the boundary of the domain with the determined dimensions.

[Figure]

**(a)**

**(b)**

[Figure]

Pressure.Gradient
Contour 1
6.0e+01
5.6e+01
5.3e+01
5.0e+01
4.6e+01
4.3e+01
4.0e+01
3.6e+01
3.3e+01
3.0e+01
2.7e+01
2.3e+01
2.0e+01
1.7e+01
1.3e+01
9.9e+00
6.6e+00
3.3e+00
0.0e+00
[kg m^-2 s^-2]

**(c)**

**Figure iii. Pressure gradient contour for U$_{ref}$ = 7 m s$^{-1}$ and(a) $\theta_{tilt} = 0^{°}$, (b) $\theta_{tilt} = 5^{°}$ (c) $\theta_{tilt} = 10^{°}$on the symmetry plane of the balloon (z=0).**

In response to the referee's query regarding the sensitivity of the computational domain size, we would like to provide additional information to further support the appropriateness of the domain size chosen in our study. In our initial response, we explained that the domain dimensions were carefully determined by evaluating flow gradients at the boundaries, to select a size that ensured zero gradients. While this method was valid, we employed sensitivity analysis of wake characteristics in relation to domain size here to further support the reliability of our approach. In light of this, we have conducted a follow-up study wherein we created two different computational domains: one with an extended upstream length (from 5d to 10d) and another with an extended downstream length (from 22.5d to 30d) as shown in figure iv relative to the turbine position, beyond the original domain utilized in the research. We increased the number of nodes in the upstream distance by a factor of 2 for the extended upstream domain and by 1.3 times (equal to the length increment ratio) for the extended downstream domain to ensure consistent spatial resolution in all cases, while only considering changes in the computational domain size on the results.

[Figure]

**Figure iv. Computational domain size for (a) extended upstream length and (b) extended downstream length.**

With these new computational domains, we applied the same simulation settings and boundary conditions as described in Section 4 of the paper to conduct simulations for the cases where $U_{ref}$ =7 m s$^{-1}$ and $\theta_{tilt} = 0°$ and $10°$. Figure v showcases the simulation results employing an extended upstream length, illustrating vertical profiles of the time-averaged normalized x-velocity at x/d = -2, within the range of -4 < y/d < 4 at z = 0. Additionally, figure vi presents the outcomes of the simulation utilizing an extended downstream length, displaying vertical profiles of the time-averaged normalized x-velocity at x/d = 5 and 14, within the range of -4 < y/d < 4 at z = 0.

[Figure]

**Figure v. Comparison of vertical profiles of the time-averaged normalized x-velocity for different upstream lengths for $U_{ref}$ =7 m s$^{-1}$ and -4<y/d<4, and z=0 with $\theta_{tilt}$= (a) $0^{\circ}$ (b) $10^{\circ}$ at x/d =-2.**

[Figure]

**Figure vi. Comparison of vertical profiles of the time-averaged normalized x-velocity for different downstream lengths for $U_{ref}$ =7 m s$^{-1}$ and -4<y/d<4, and z=0 with $\theta_{tilt}$= $0^{\circ}$ at (a) x/d =5 (b) x/d= 14 and $\theta_{tilt}$= $10^{\circ}$ at (c) x/d =5 (d) x/d= 14.**

According to figure v, extending upstream lengths has been observed to have a relatively minor impact on the velocity profiles at various locations relative to the wind turbine position. This observation can be ascribed to the specific conditions governing our investigation. Notably, we considered our wind turbine situated at a high altitude where the atmosphere tends to be more stable because it is less influenced by surface heating and friction, which can lead to reduced turbulence and vertical mixing. The absence of significant boundary layer effects due to the high-altitude location of our wind turbine led to a longer upstream length less critical for capturing boundary layer-related phenomena. These specific environmental conditions enabled us to design our computational domain with a smaller upstream length than typically required for studying ground-based wind turbine wake behaviour, prioritizing computational efficiency while maintaining result accuracy.

Furthermore, as shown in figure vi, prolonging the downstream distances has shown only a marginal influence on the velocity profiles at different positions relative to the wind turbine's location. Our choice of downstream length was carefully considered in light of several critical factors. The selected downstream domain size was designed to encompass the essential characteristics of the wake, including wake recovery, turbulence decay, and gradual mixing with ambient air. This careful consideration of downstream length was paramount to accurately capturing the wake's behaviour and its impact on downstream flow. In summary, under these controlled conditions and with careful attention to factors critical to wake simulation, the impact of extending the domain size upstream and downstream remained minimal, providing robust support for the appropriateness of our chosen computational domain size.

5. Validation studies use conventional wind turbines, while the case being studied is much more complex.

Due to the novelty of balloon wind turbines, there is a lack of experimental data in the literature to validate the numerical results pertaining to their aerodynamics and wake flow. Nevertheless, it is possible to assess the methodology employed in this study to characterize the wake flow of these turbines. To this end, we employed the same methodology to investigate the wake behaviour of a smaller turbine that had been previously studied experimentally.

There are two key differences between the balloon wind turbine and the smaller turbine: the diameter of their rotors and the presence of the balloon in the main model. Based on the results obtained from the balloon wind turbine, it is evident that the effective length of the rotor wake is considerably longer than that of the balloon, and they do not interact under various inflow conditions considered in this study. Hence, the wake flow behaviour of the rotor is almost independent of the balloon.

Although the turbine being studied differs in diameter from the smaller turbine used for validation, it is still significant to validate our methodology using a smaller turbine with available experimental data. The purpose of this validation was to demonstrate the accuracy and reliability of the numerical approach, irrespective of the specific turbine size or complexity.

Furthermore, the utilization of LES enables high-resolution analysis of the turbulent flow structures within the turbine wake. By adequately resolving the flow features, our model captures

the intricate complexities associated with different turbine sizes. The results obtained from the validation study showcased a satisfactory agreement between the LES-ADM method and experimental measurements. Therefore, in the absence of experimental data for the balloon wind turbine, we argue that validating the robustness of our methodology through promising outcomes obtained using the same approach serves as a reliable indicator of the accuracy of the main model's findings.

6. If the authors claim to use LES, is the energy in the proximity of the balloon wall resolved well?

In LES, the $y^+$ values should be lower than 5 for wall-resolved LES simulations. Moreover, the Dynamic Smagorinsky model tends to perform better in capturing near-wall turbulence compared to traditional static SGS models, as it adapts the SGS model coefficients dynamically based on the local flow characteristics. When using this model, a lower y Plus value (closer to 1) is often recommended for accurate resolution of the near-wall turbulent structures. This is because the model is designed to capture small-scale motions more effectively, and a finer resolution near the wall helps to capture the important energy-containing turbulent eddies. When the $y^+$ value falls within this range, it generally indicates that the energy near the wall is resolved well. To reach these values near the balloon wall in the simulations, the mesh size and its spacing near the wall were adjusted accordingly. Figure vii illustrates $y^+$ contours for $U_{\text{ref}} = 7$ m s$^{-1}$ and $\theta_{tilt} = 0°, 5°$ and $10°$ on the balloon wall. According to the figure, the value of this parameter is close to 1 in most regions on the wall and does not exceed 3.5 in any region for different wind scenarios.

[Figure]

(a)

[Figure]

(b)

(c)

**Figure vii. Wall Yplus contours for $U_{ref} = 7$ m s$^{-1}$ and (a) $\theta_{tilt} = 0°$, (b) $\theta_{tilt} = 5°$ (c) $\theta_{tilt} = 10°$ on the balloon's wall.**

7. The authors focus the studies on the wake, but the mesh behind the balloon does not seem to be well refined such that it can well resolve small eddies. Perhaps plot the Q or Lambda2 criterion?

To determine the mesh distribution in the wake region, the LES-friendly mesh generation algorithm, which was described in section 5.1 of the manuscript, was adopted. Evidence confirming the reliability and suitability of the mesh generation approach has been provided in response to the third query posed by Referee #2. To further investigate the mesh size in the region of interest, the mesh distribution in the near-wake is illustrated in figure viii (a). To resolve small eddies in the wake of the turbine and the vicinity of the balloon's separation region (close to the trailing edge of the balloon's airfoil), the mesh resolution in these areas

was significantly enhanced compared to other regions within the computational domain, as illustrated in figure viii (a). The magnitude of Lambda 2 for the case where $U_{ref}$ =7 m s$^{-1}$ and $\theta_{tilt} = 0°$ in the symmetry plane of the balloon (z=0) is clipped for small eddies and the corresponding contour is depicted in figure viii (b). According to figure viii (b), the small eddies in the separation zone of the balloon and the wake of the turbine are properly resolved.

[Figure]

(a)

(b)

Figure viii. (a) Mesh distribution (b) Lambda 2 contour on the symmetry plane of the balloon

8. Last minor aspect: remove the 6th point in Conclusion as nothing is written there.

Thank you for pointing out this error; we have removed it.

---

## Author Response (AR1)

Dear associate editor,

Firstly, we would like to express our gratitude to the referees for their valuable feedback and suggestions on our manuscript. We have carefully considered each comment and responded to them point-by-point. In addition, we made corresponding revisions to improve the quality of our paper. We appreciate the opportunity to revise and resubmit our manuscript to the Wind Energy Science Journal.

Sincerely,

Dr. Mostafa Varmazyar
Associate Professor
Shahid Rajaee Teacher Training University

---

## Author Response (AR2)

Dear  Dr. Bianchini,

We want to express our sincere gratitude for your dedicated support and the efforts of the referees in improving our paper. In response to their feedback, we diligently revised our previous responses to the reviewers and made corresponding revisions, addressing all concerns and enhancing their clarity. Additionally, as it appeared that our previous response to Referee #2 was not readily available in the interactive discussion, we have modified and uploaded our responses to Referee #2 and other referees in the interactive discussion section as final author comments. Your guidance and the thorough review process have been invaluable, and we hope that these modifications effectively address the reviewers' suggestions and contribute to the paper's overall improvement.

Sincerely,

Dr. Mostafa Varmazyar
Associate Professor
Shahid Rajaee Teacher Training University

---

## Author Response (AR3)

Dear Dr. Bianchini,

We wish to extend our sincere appreciation for your dedicated efforts in guiding our paper through the review process, which has significantly enhanced its quality. We are also deeply grateful to your colleagues for their invaluable contributions throughout the submission process, and we appreciate the Handling Chief Editor's generous decision to waive the APC, enabling us to share our research in this esteemed journal. Additionally, we would like to express our gratitude to the referees for their valuable comments, which have enriched our paper. In response to your and the esteemed reviewers' suggestions, we have incorporated the description of the independence of the results from the chosen domain size into Line 225 of the revised manuscript. We hope this modification effectively addresses your and the reviewers' recommendations.

Sincerely,

Dr. Mostafa Varmazyar
Associate Professor
Shahid Rajaee Teacher Training University

---

## Author Response (AR4)

We would like to extend our sincere gratitude to respected Associate Editor Dr. Bianchini and respected Chief Handling Editor Dr. Veers for their invaluable guidance and meticulous attention, which significantly contributed to the paper's improvement. We also want to express our profound thanks to the entire team at the WES for helping our paper for publication. We are profoundly thankful to the diligent reviewers for their insightful feedback and constructive criticism, which greatly enhanced the quality of our research. Being published in the WES, a renowned platform for high-quality research, is a privilege, and we are eager to share our findings with the global scientific community. Thank you to all involved for your commitment to scientific excellence; we look forward to further collaborations with the journal in the future.

Sincerely,

Aref Ehteshami

Shahid Rajaee Teacher Training University